# YeeE-like bacterial SoxT proteins mediate sulfur import for oxidation and signal transduction
Jingjing Li [1], Fabienne Göbel [1,2], Hsun Yun Hsu[1], Julian Nikolaus Koch[1,3], Natalie Hager[1], Wanda Antonia Flegler[1,4], Tomohisa Sebastian Tanabe [1,5] & Christiane Dahl [1] ✉

Many sulfur-oxidizing prokaryotes oxidize sulfur compounds through a combination of initial extracytoplasmic and downstream cytoplasmic reactions. Facultative sulfur oxidizers adjust transcription to sulfur availability. While sulfur-oxidizing enzymes and transcriptional repressors have been extensively studied, sulfur import into the cytoplasm and how regulators sense external sulfur are poorly understood. Addressing this gap, we show that SoxT1A and SoxT1B, which resemble YeeE/YedE-family thiosulfate transporters and are encoded alongside sulfur oxidation and transcriptional regulation genes, fulfill these roles in the Alphaproteobacterium *Hyphomicrobium denitrificans*. SoxT1A mutants are sulfur oxidation-negative despite high transcription levels of sulfur oxidation genes, showing that SoxT1A delivers sulfur to the cytoplasm for its further oxidation. SoxT1B serves as a signal transduction unit for the transcriptional repressor SoxR, as SoxT1B mutants are sulfur oxidation-negative due to low transcription unless SoxR is also absent. Thus, SoxT1A and SoxT1B play essential but distinct roles in oxidative sulfur metabolism and its regulation.

The biogeochemical cycle of sulfur is primarily driven by prokaryotes, which reduce sulfate or sulfite in an anaerobic respiratory process to conserve energy[1]. Dissimilatory sulfur oxidizers maintain the cycle by oxidizing reduced sulfur compounds and using them as electron donors for energy conservation through respiration or photosynthesis[2,3]. Sulfide and thiosulfate ($S_2O_3^{2-}$) are common sulfur substrates in these organisms and in many cases their oxidation is initiated outside of the cytoplasm (if present in the bacterial periplasm). Further oxidative steps take place in the cytoplasm. This requires the import of sulfur into this cellular compartment[2,4–6]. In organisms that use reduced sulfur compounds as alternative or additional electron donors to organic compounds, transcriptional regulation of sulfur oxidation allows adaptation of metabolic flux to environmental conditions[7,8]. Sulfur transport across the cytoplasmic membrane is likely involved in the sensing and response to externally available reduced sulfur compounds. While intensive experimental work has been dedicated to elucidating the wide variety of redox reactions involved in prokaryotic sulfur oxidation[2,3], less effort has been devoted to clarifying the mechanisms of sulfur transport required for its use as an electron source or in the course of signal transduction. Uptake of sulfur compounds for assimilatory purposes, i.e., for the biosynthesis of sulfur-containing cell constituents, has been

much better investigated and provides starting points for answering the many open questions.

Assimilation of sulfur is required for the growth of all living beings and prokaryotes obtain it either from inorganic sulfate or from organosulfur compounds such as sulfonates, sulfate esters, or sulfur-containing amino acids[9–13]. Transporters mediating the import of such precursors include a variety of ABC-type systems with solute-binding proteins as the primary determinants of transporter specificity[11]. The *Escherichia coli* CysUWA complex is a prime example for this concept. It takes up sulfate and thiosulfate as a sulfur source and acts in combination with periplasmic Sbp and CysP, respectively[9,12]. Recent work has shown that *E. coli* has an additional transporter, TsuA, which imports thiosulfate as a source of sulfur[14–16]. The protein belongs to the YeeE/YedE family (COG2391; DUF395) and has nine transmembrane helices. The structurally characterized protein from *Spirochaeta thermophila* contains three conserved cysteine residues that play a role in transport, probably through transient hydrogen bond mediated interaction with thiosulfate ions[14]. In *E. coli*, the soluble cytoplasmic protein TsuB (YeeD), that is encoded immediately adjacent to *tsuA*, is essential for thiosulfate uptake via TsuA[15,16]. TsuB is similar to, but cannot replace, TusA[17], which is a central sulfur hub in bacterial cells[18]. In the archaeon

---

[1]Institut für Mikrobiologie & Biotechnologie, Rheinische Friedrich-Wilhelms-Universität Bonn, Bonn, Germany. [2]Present address: Institute for Integrated Natural Sciences, University of Koblenz, Koblenz, Germany. [3]Present address: Department of Biochemistry, Institute of Biosciences, University of Rostock, Rostock, Germany. [4]Present address: Institut für Ernährungs- und Lebensmittelwissenschaften, Rheinische Friedrich-Wilhelms-Universität Bonn, Bonn, Germany. [5]Present address: Division of Microbial Ecology, University of Vienna, Vienna, Austria. ✉e-mail: ChDahl@uni-bonn.de

*Methanococcus maripaludis* a YedE-like protein is involved in transport of selenium, which is chemically similar to sulfur[19,20]. PmpA and PmpB from *Serratia* sp. ATCC39006 are other members of the YeeE/YedE family that have been predicted to transport sulfur-containing ions, albeit not for assimilatory purposes[21]. Similar proteins facilitate the uptake of extracellular zero-valent sulfur across the cytoplasmic membrane, thereby increasing cellular sulfane sulfur levels in bacterial cells[22].

Genes encoding YeeE/YedE-like proteins also occur together with genes for sulfur-metabolizing enzymes in sulfur-oxidizing prokaryotes. In the Alphaproteobacteria *Paracoccus pantotrophus* GB17[T] (DSM2944[T]), *Pseudaminobacter salicylatoxidans* KCT001, and *Hyphomicrobium denitrificans* X[T] (DSM 1869[T]), large *sox* gene clusters encoding the thiosulfate-oxidizing periplasmic Sox multienzyme system are accompanied by *soxT* genes encoding YeeE-like transporters[8,23–25]. In these organisms, *soxT* is located in a *soxSRT* arrangement. SoxR, a repressor protein, binds to the promoter-operator region of the *sox* operon and prevents transcription when sulfur compounds are absent[8,23,24]. This suggests a potential role in signal transduction for the membrane protein. In *Paracoccus denitrificans* PD1222 (DSM 104981), *Cereibacter sphaeroides* (formely *Rhodobacter sphaeroides*[26]), and *Roseovarius* sp. 217, the *sox* genes are flanked by two *soxT* genes[27]. We denote the one in the *soxRST* arrangement *soxT1* and term the other *soxT2*.

Two *soxT* genes are also found in *H. denitrificans*. This genetically tractable bacterium serves as a model for the elucidation of the cytoplasmic sulfur-oxidizing sHdr-LpbA pathway[5,7,28–31]. In *H. denitrificans*, thiosulfate oxidation starts in the periplasm, where the SoxXAB proteins work together to oxidatively conjugate thiosulfate to a conserved cysteine of the substrate-binding protein SoxYZ and release a sulfate molecule[2,7,32,33]. The second sulfur atom of the original thiosulfate molecule is by unknown means transferred to the cytoplasm, where it is oxidized to sulfite by the sHdr-LbpA system[5,6]. In *H. denitrificans*, the typical *soxSRT* arrangement resides immediately upstream of the genes for a TusA-like sulfur carrier protein and a putative cytochrome P450[7,8]. A second *soxT* gene is located downstream of the large set of genes that encode the enzymes for cytoplasmic sulfite formation and is transcribed divergently from them.

Here, we set out to decipher the function of the two different potential SoxT transporters in *H. denitrificans*. To this end, we collected information on the distribution and phylogeny of related transporters in sulfur-oxidizing prokaryotes and constructed a set of informative mutant strains lacking the transporter genes, the genes for two different transcriptional regulators, *soxR* and *shdrR*, and combinations thereof. Phenotypic characterization of the mutants and comparative analysis of transcription levels for relevant sulfur-oxidizing proteins finally allow functional assignments.

## Results

### Occurrence and phylogeny of YeeE/YedE-like proteins

Members of the YeeE/YedD family are found in organisms across a wide variety of metabolic pathways and prokaryotic phyla, both within the Archaea and the Bacteria[14,21,22,28]. As of March 2022, the Database of Clusters of Orthologous groups included complete genomes from 1187 bacteria and 122 archaea. Among the latter, YeeE-type proteins occur in *Saccharolobus* and *Sulfolobus* (Thermoproteota) and some representatives from the Thermoplasmatota. Among the bacteria, some YeeE-containing representatives are found in the phyla Actinobacteriota, Bacteroidota, Cyanobacteriota, Deinococcota, Bacillota, Spirochaetota, Verrucomicrobiota and Thermotogota, while there are many organisms with YeeE among the Pseudomonadota and the Desulfobacterota.

Conspicuously, the proteins of the YeeE family vary greatly in length. The structurally characterized *S. termophila* TsuA and relatives, as well as the SoxT proteins, share lengths of 330 to 350 aa, nine transmembrane helices and three conserved cysteines. In contrast, PmpA and PmpB, as well as their relatives[21,22], are much shorter, approximately 130 amino acids in length. They share four predicted transmembrane helices and one conserved cysteine residue. We re-evaluated the relationship between the long and short members of the family and found that PmpB and related proteins

align perfectly with the N-terminal half of the full-length YeeE family members, while PmpA and relatives match with their carboxy-terminal half (Supplementary Fig. 1). PmpB contains one cysteine that is in the same position as the second conserved cysteine of *S. termophila* TsuA (Cys[91]) and a PmpA cysteine matches the third conserved cysteine (Cys[293]). Cys[91] and Cys[293] are indispensable for proper function of the *S. termophila* transporter[14]. The central transmembrane helix (H7 in *S. thermophila* TsuA) is not covered by the PmpAB sequences. We propose that PmpA and PmpB form a heterodimer and that together they perform functions similar to those of the YeeE proteins. The similarity of PmpA to PmpB suggests that they arose from a gene duplication. The two genes may then have fused and acquired an element encoding an additional transmembrane helix, resulting in the full-length YeeE family proteins.

As a first step towards a sequence-based grouping of YeeE-like proteins from dissimilatory sulfur-oxidizing bacteria, we created a phylogenetic tree including all YeeE-like transporters encoded in organisms containing the full set of *soxXABYZ* genes. The functionally characterized TsuA transporters from *E. coli* and *S. thermophila* were also included (Fig. 1). The tree reveals multiple paralogous groups with the TsuA proteins residing on a well separated branch. The most closely related group consists of SoxT2 proteins such as those encoded in close proximity to the *sox* genes in *C. sphaeroides*, *P. denitrificans* PD1222, and *P. salicylatoxydans*. SoxT1 proteins form another coherent clade, distant from the SoxT2 group. Both *soxT* genes from *H. denitrificans* are of the SoxT1 type and we term them SoxT1A (Hden_0681) and SoxT1B (Hden_0699).

Further information was obtained by screening all sulfur-oxidizing prokaryotes containing *shdr* genes for the presence of *soxT1*, *soxT2*, *sox* genes, genes for the sHdr-LbpA sulfur-oxidizing system and genes for the transcriptional repressors sHdrR[7] and SoxR[8] by HMSS2[34]. Clusters of genes encoding the sHdr-LbpA pathway for sulfane sulfur oxidation in the cytoplasm fall into two distinct categories. Type I and type II sHdr systems share the Fe/S flavoprotein sHdrA, the electron carrier protein sHdrC1 and the proposed catalytic subunit sHdrB1. The type I sHdrC2 and sHdrB2 polypeptides are encoded by a fused gene, *shdrB3* in the type II-containing organisms[5,6]. As evident from Fig. 2, SoxT transporters are only rarely present in genomes with the type I *shdr* genes and completely absent in genomes with type II sHdr, even though some of these organisms harbor the capacity for Sox-driven thiosulfate oxidation (see also Supplementary Data 1). In all genomes encoding the regulator SoxR, either SoxT1 or SoxT2 is present. The same is not true for the related repressor sHdrR. It does not always co-occur with SoxT1 or SoxT2. This is in line with a possible function for SoxT1 and/or SoxT2 in SoxR-dependent gene regulation, but contradicts a general role for the transporters in sulfur compound import. Nevertheless, sulfur import may be facilitated by either one of the transporters in a subset of sulfur oxidizers.

### Regulation of *yeeE*-like genes in *Hyphomicrobium denitrificans*

*H. denitrificans* has the genetic potential for two distinct pathways of thiosulfate oxidation, both of which occur or begin in the periplasm. At 5 mM, all thiosulfate is converted to tetrathionate[28]. This reaction is catalyzed by thiosulfate dehydrogenase (TsdA), a periplasmic diheme cytochrome *c*. At 2.5 mM or less, most thiosulfate is oxidized to sulfite and eventually to sulfate. However, the formation of tetrathionate is not completely suppressed in wildtype *H. denitrificans* even at low substrate concentrations. This makes phenotypic characterization of mutant strains difficult[28,29]. Strains lacking thiosulfate dehydrogenase (ΔtsdA) are unable to form tetrathionate and are ideal for studying the processes involved in the complete oxidation of the substrate to sulfate[6–8,30]. Therefore, all experiments reported in this and also our previous studies on the transcriptional regulation of sulfur oxidation[7,8] were performed with a *H. denitrificans* ΔtsdA reference strain.

RT-qPCR provided initial evidence that SoxT1A and SoxT1B from *H. denitrificans* may be intricate components of the oxidation pathway and/or involved in its transcriptional regulation[8]. The transcript abundance for *soxT1A* increased more than tenfold upon addition of thiosulfate in the

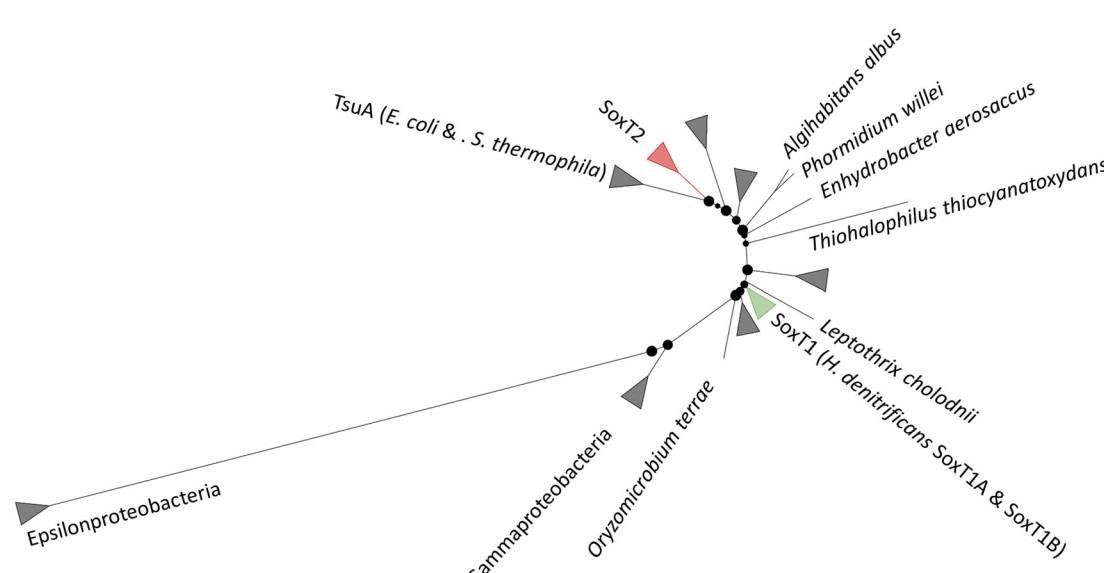

Tree scale: 1

**Fig. 1 | Unrooted phylogenetic tree for YeeE-like proteins in Sox-containing sulfur oxidizers.** Groups of proteins encoded in or immediately adjacent to *sox* clusters (SoxT1 and SoxT2) are highlighted. SoxT1 includes both studied transporters from *H. denitrificans*. The TsuA group includes the thiosulfate uptake proteins from *E. coli* and *S. thermophila*[14,16]. The tree was calculated with 1000 bootstrap resamplings using Ultrafast Bootstrap[55] and IQ-Tree[57,58]. Bootstrap values between 50% and 100% are displayed as scaled circles at the branching points. Protein accession numbers and species names are available at https://github.com/WandaFlegler/Masterarbeit/blob/main/Galaxy3-%5BBMGE_Cleaned_sequences_Fasta%5D.fasta.treefile.

*H. denitrificans* Δ*tsdA* reference strain, while *soxT1B* expression remained essentially unaffected and thus similar to the expression of the genes for the transcriptional repressors *soxR* and *shdrR*[8]. Here, we extend these analyses with genome-wide mRNA-Seq data for the reference strain, comparing transcription in the absence and presence of 2 mM thiosulfate. Of the 3529 predicted genes, 3379 mRNAs (95.7%) were identified. The availability of thiosulfate affected the abundance of a total of 136 (4.1%) of the detected mRNAs (Supplementary Fig. 2, Supplementary Tables 1 and 2). In the presence of thiosulfate, mRNA transcripts of 47 genes showed lower mRNA abundance than in its presence (Supplementary Table 1), among them several genes for enzymes of fatty acid biosynthesis (acyl carrier protein (ACP), β-hydroxyacyl-ACP dehydratase, β-ketoacyl-ACP synthase) and assimilatory sulfate reduction (assimilatory sulfite reductase, sulfate adenylyltransferase).

mRNA transcripts of 89 genes (18 genes for hypothetical proteins) were more abundant in the presence of thiosulfate (Supplementary Table 2). The most affected gene (Hden_0834, YeiH) with a fold change of +526 (Supplementary Fig. 3a) encodes a putative efflux pump belonging to the PSE (Putative Sulfate Exporter) family (entry 2.A.98, Transporter Classification Database). The classification as a putative sulfate exporter is based on a study in *Paracoccus pantotrophus*, where 3-sulfolactate is converted to pyruvate and sulfite during the dissimilation of cysteate. It has been proposed that sulfite is oxidized to sulfate in the cytoplasm and then exported by the transporter, designated SuyZ in *P. pantotrophus*[35]. However, sulfite dehydrogenases of *Paracoccus* species are periplasmic enzymes and it is more likely that the transporter extrudes sulfite from the cytoplasm into the periplasm where it is then detoxified by oxidation to sulfate. A similar role may be played by YeiH in *H. denitrificans*, which not only closely resembles *P. pantotrophus* SuyZ at the amino acid sequence level (30% identity, 56.4% similarity), but also shares the same structural features (Supplementary Fig. 4). The second most affected gene (Hden_0835) encodes a LysR-type transcriptional activator and resides next to *yeiH* (Supplementary Table 3 and Supplementary Fig. 3a). Strong increases, up to 20-fold, were also observed for the transcripts from the *shdr-lbpA2-sox* locus. Those for *soxT1A* were among the top three (Fig. 3a). In full agreement with RT-

qPCR analysis (Fig. 3b,c), the transcription of only three genes in the genomic sulfur oxidation region shown in Fig. 3a proved unaffected in the mRNA-Seq experiment, and these were the genes for the two transcriptional repressors, sHdrR and SoxR, and *soxT1B*. These results are consistent with previously published RT-qPCR analyses showing that transcription of these genes was barely affected by thiosulfate[8]. We thus state with confidence that in the Δ*tsdA* reference strain *soxT1A* expression increases substantially during thiosulfate oxidation, while *soxT1B* expression along with that of *soxR* and also *shdrR* does not change significantly. These findings are corroborated by RT-qPCR analysis for *H. denitrificans* strains Δ*tsdA* Δ*soxR*[8] and Δ*tsdA* Δ*shdrR*[7], which lack the individual repressor genes. In the repressor-negative strains, *soxT1A* expression is high even in the absence of thiosulfate, while *soxT1B* transcript abundance is not affected (Fig. 3b). When the strains deficient in one or the other transcriptional repressor are grown in the presence of thiosulfate, transcript abundance for *soxT1A* along with that for *shdrA* raises substantially above the level observed in the absence of the sulfur compound (Fig. 3c). This can be explained by the action of the remaining regulator protein in these strains, which requires the presence of an oxidizable substrate for full release of transcriptional repression. Note that the increase in transcript abundance for *soxT1B* was almost negligible (from 1.2 to 4.5 fold in the Δ*tsdA* Δ*soxR* and from 0.6 to 2.8 fold in the Δ*tsdA* Δ*shdrR* strain). Also note that the transcript levels for the tested genes are lower in the Δ*tsdA* Δ*shdrR* strain in the absence of thiosulfate (Fig. 3b) than in the *H. denitrificans* Δ*tsdA* reference strain in the presence of the sulfur substrate (Fig. 3c). This observation cannot be fully explained yet but is certainly related to the second regulator SoxR. This protein is still present in the Δ*tsdA* Δ*shdr* strain, and transcriptional repression is apparently not fully relieved by the absence of sHdrR alone. Experiments described further below confirm SoxR as the major regulator of sulfur oxidation in *H. denitrificans*. Clearly, the interplay of the two regulators needs to be investigated in more detail in the future.

Our mRNA-Seq analyses also yielded insight into the transcription of further *yeeE*-like genes in *H. denitrificans*, i.e., *pmpA* and *pmpB*. In contrast to *soxT1A*, the expression of *pmpA* and *pmpB* is not affected by the availability of thiosulfate (Supplementary Fig. 5). These genes attracted our

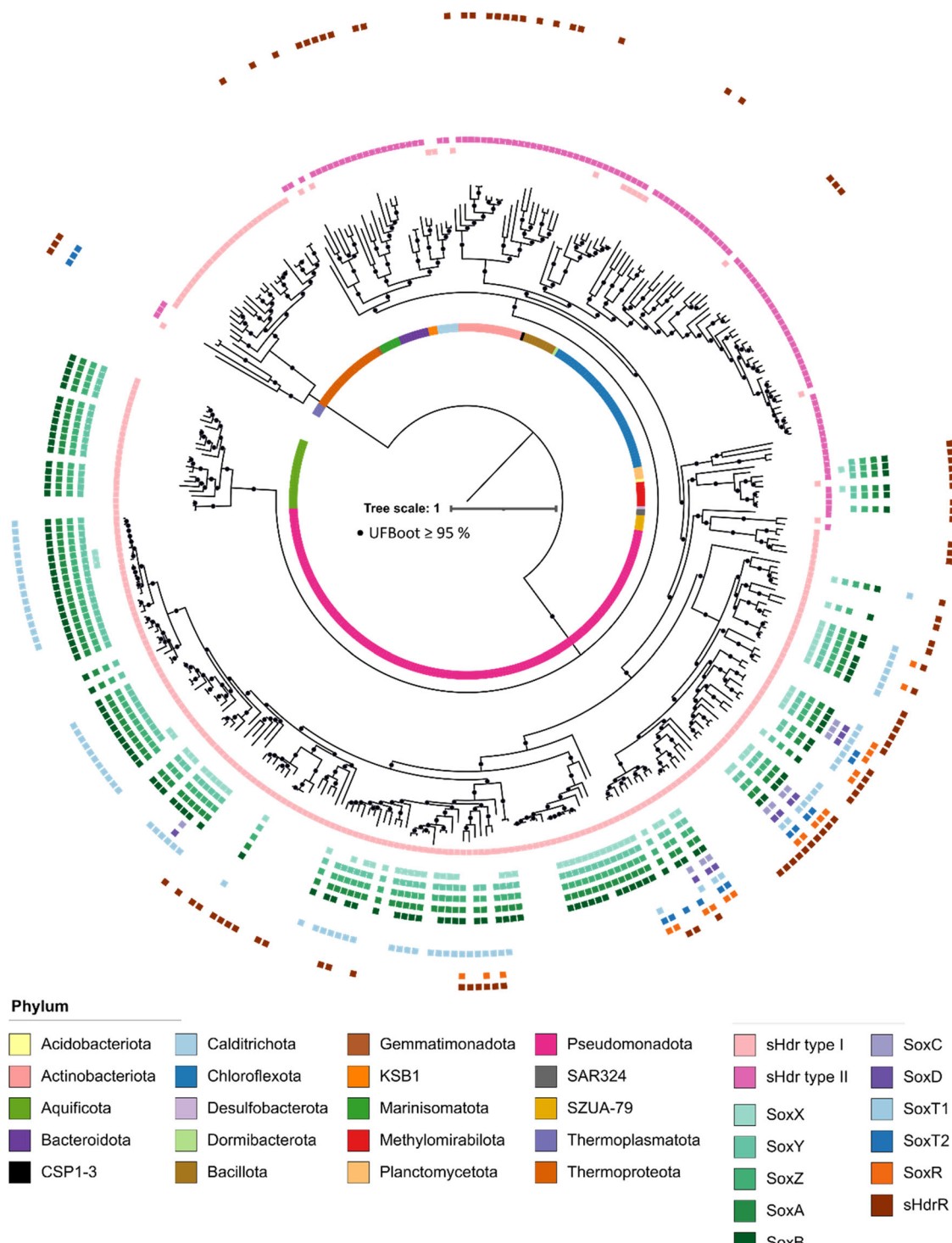

**Fig. 2 | Distribution of SoxT transporters in sulfur oxidizers with sHdr systems.**
The distribution of genes for type I and type II sHdr systems, full Sox systems
enabling complete oxidation of thiosulfate in the periplasm (SoxAXBYZCD),
truncated Sox systems requiring formation of sulfite in the cytoplasm (SoxXABYZ),
SoxT and two different transcriptional repressors, SoxR and sHdrR, is shown. In
order to be classified as present, at least the proteins SoxA, SoxB, SoxY, and SoxZ had
to be encoded in a syntenic gene cluster. The *sox* genes of species of the order
Ectothiorhodospirales are not syntenic and were therefore assigned manually as
described before[59]. A type I sHdr system was marked positive when the core genes
*shdrC1B1AHC2B2* were present in a syntenic gene cluster. For assignment of a type
II sHdr system, *shdrC1B1AHB3* and *etfAB* had to be present in a single syntenic gene
cluster[6] The gene for the regulator sHdrR was only considered positive when located
in or immediately next to a *shdr* gene cluster. The species tree was calculated as
described before[6]. The data underlying the figure is provided in Supplemen-
tary Data 1.

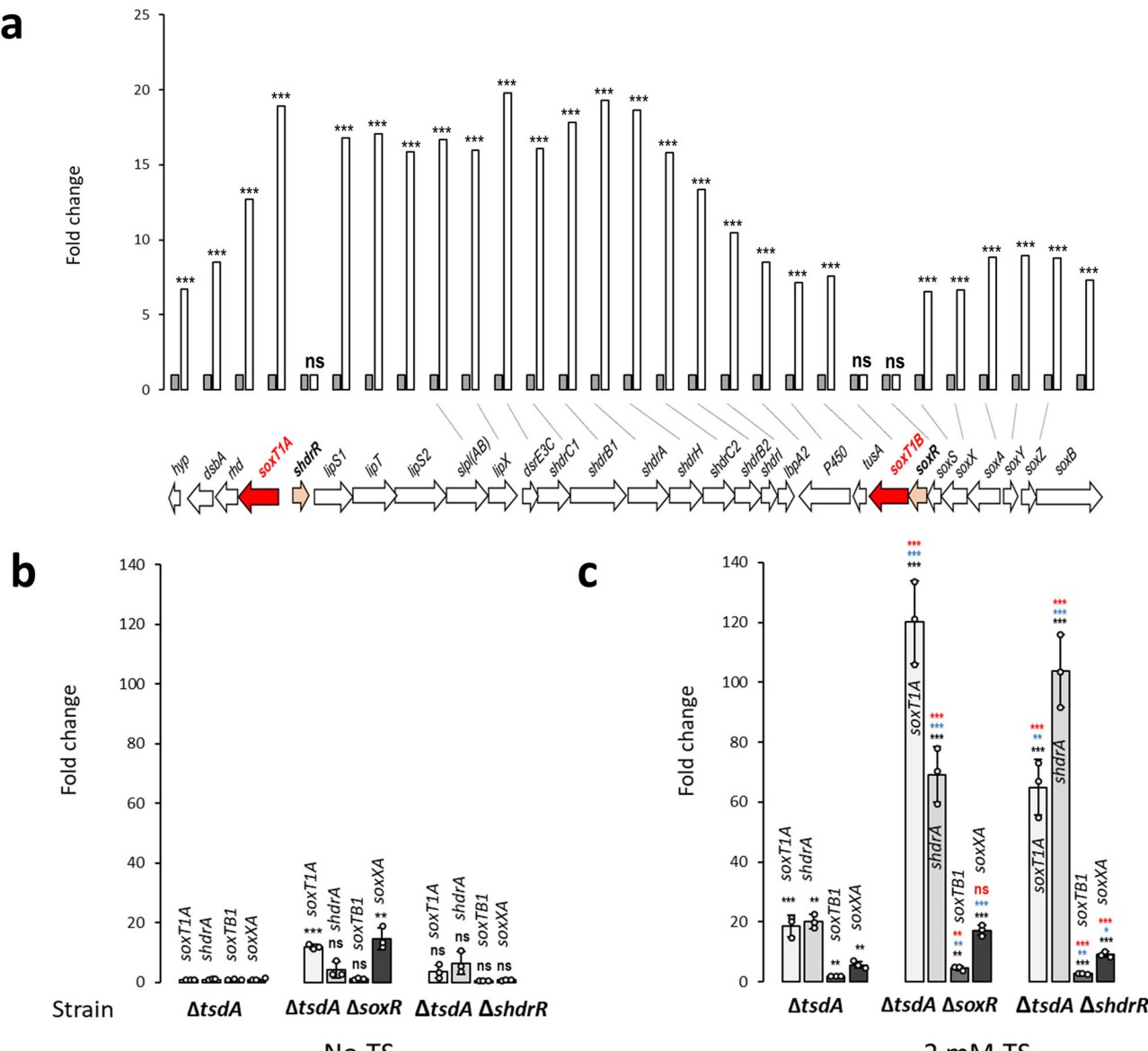

**Fig. 3 | Thiosulfate-dependent regulation of gene expression in the *lip-shdr-lbpA-sox* locus in *H. denitrificans*. a** Transcript abundance changes of genes encoding enzymes involved in thiosulfate oxidation in the *H. denitrificans* Δ*tsdA* reference strain as assessed by mRNA-Seq analysis. Columns in gray show the reference values for cells grown in the absence of thiosulfate, white columns apply to cells grown with 2 mM thiosulfate. The experiment was conducted in duplicate, each time using mRNA preparations from two different cultures. Adjusted *p* values for statistically significant changes were all below 0.001 (Supplementary Table 2) and are indicated by three asterisks (*** *p* < 0.001); ns, not significant. Relative mRNA levels of four indicative genes/combinations of genes located in the *shdr-sox* genetic region (depicted in **a**) from *H. denitrificans* for the Δ*tsdA* reference strain and the regulator deficient strains Δ*tsdA* Δ*soxR* and Δ*tsdA* Δ*shdrR* grown in the absence (**b**) or in the

presence of 2 mM thiosulfate (**c**). All changes are compared to *H. denitrificans* Δ*tsdA* in the absence of thiosulfate. Results were adjusted using *H. denitrificans rpoB*, which encodes the β-subunit of RNA polymerase, according to[60]. All three strains grow equally well on methanol and are capable of thiosulfate oxidation[7,8]. Three parallel experiments were performed to obtain the averages and standard deviation. Data are presented as means ± SD. Individual data points are indicated. One-way ANOVA was performed to calculate *p*-values. Asterisks indicate statistically significant differences (**p* < 0.05, ***p* < 0.01, ****p* < 0.001, ns = (*p* > 0.05)). Color coding: black: differences compared to strain Δ*tsdA* in the absence of thiosulfate; blue, differences compared to Δ*tsdA* in the presence of thiosulfate; red, differences compared to the same strain in the absence of thiosulfate. TS thiosulfate.

attention because they are located in close proximity to genes encoding proteins that may be related to sulfur metabolism, such as a Sox(YZ) fusion and SoxH[36]. However, close inspection revealed that the gene ensemble rather encodes a PQQ-dependent enzyme for alcohol catabolism, its electron acceptor and an associated transport system (Supplementary Fig. 5). Since there is no evidence that PmpAB is involved in transport processes relevant to oxidative sulfur metabolism in *H. denitrificans*, they were not analyzed further.

## Role of SoxT1B in *H. denitrificans*: gene inactivation, complementation, and cysteine exchanges

To clarify the role of SoxT1B, an *H. denitrificans* strain carrying an *in frame* deletion of the gene was constructed and phenotypically characterized. In addition, a complemented strain was investigated. Both strains grew equally well on methanol in the absence of thiosulfate (Supplementary Fig. 6a). While the deletion mutant proved negative with regard to thiosulfate oxidation, the complemented strain *H. denitrificans* Δ*tsdA soxT1Bcomp*

**Fig. 4 | Growth and thiosulfate consumption for the *H. denitrificans* Δ*tsdA* reference strain compared with strains lacking *soxT1B* or producing SoxT1B with cysteine to serine exchanges.** Thiosulfate consumption (**a**) and growth (**b**) of *H. denitrificans* Δ*tsdA* (black), Δ*tsdA* Δ*soxT1B* (bright red), Δ*tsdA* *soxT1Bcomp* (gray), Δ*tsdA* *soxT1B* $C^{24}S$ (orange), Δ*tsdA* *soxT1B* $C^{98}S$ (pink) and Δ*tsdA* *soxT1B* $C^{304}S$ (dark red). All strains were grown in 48-well microtiter plates as previously described[7] in medium containing 24.4 mM methanol and 2 mM thiosulfate. Precultures contained 2 mM thiosulfate. In (**a**), data were measured using $n = 3$ experiments and are presented with the individual measurements (small symbols) and as the mean value of these measurements ± SD (big symbols). In (**b**), representative experiments of biological duplicates are shown. Source data are provided as Source Data 2.

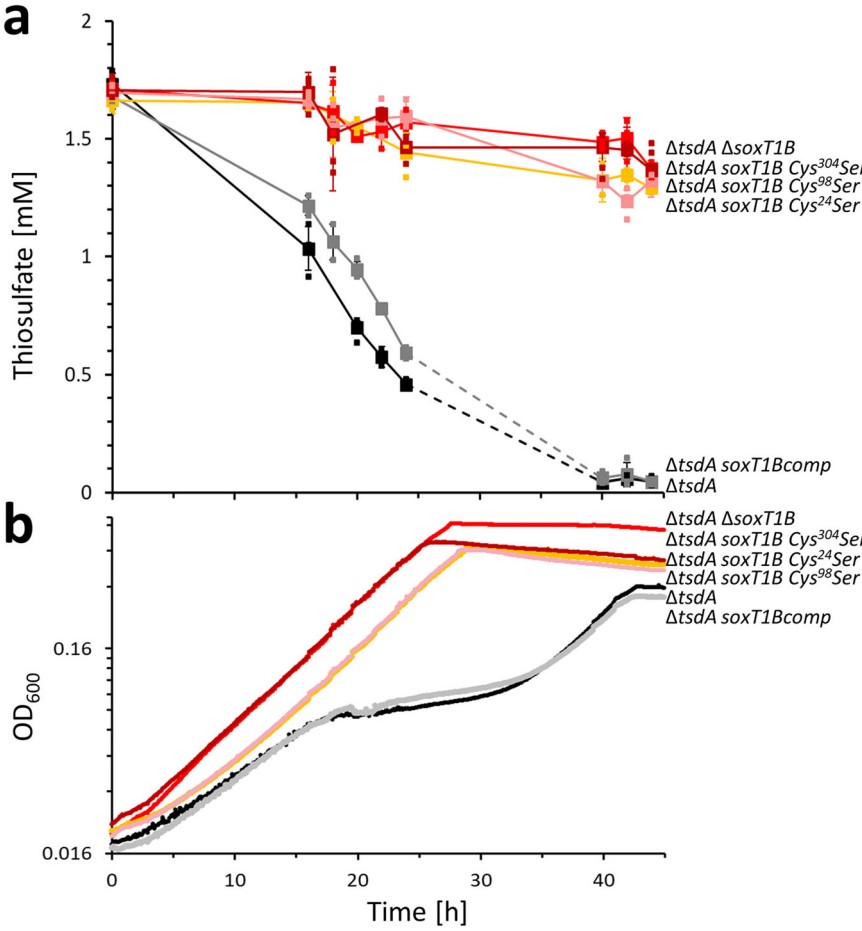

oxidized thiosulfate with the same rate as the reference strain (Fig. 4a). When the *H. denitrificans* Δ*tsdA* reference strain is grown with thiosulfate as an additional electron source, it excretes toxic sulfite[7], which causes growth retardation (Fig. 4b). In line with the thiosulfate oxidation capabilities of the studied strains, growth retardation was not observed for *H. denitrificans* strain Δ*tsdA* Δ*soxT1B* and returned upon complementation *in cis* of the Δ*soxT1B* deletion strain with an intact copy of the *soxT1B* gene (Fig. 4b).

As a proof of principle, the three cysteine residues conserved in the YeeE and SoxT proteins (Supplementary Fig. 1) were individually replaced by serine through site directed mutagenesis of the chromosomal *H. denitrificans soxT1B* gene. All three strains encoding variants of SoxT1B with cysteine to serine substitutions showed no growth retardation in the presence of thiosulfate and were unable to oxidize the sulfur compound, confirming the essentiality of these residues (Fig. 4).

### Role of SoxT1B in *H. denitrificans*: interaction with transcriptional regulators

In principle, the thiosulfate oxidation-negative phenotype of the *H. denitrificans* Δ*tsdA* Δ*soxT1B* strain can be explained by two fundamentally different functions of the membrane protein: (1) Either it is essential for import of oxidizable sulfur into the cytoplasm or (2) it is essential for signal transduction, informing one or both transcriptional repressors about the presence of external thiosulfate. In the latter case, simultaneous removal of the genes for the signal transducing membrane protein and the transcriptional repressor should allow thiosulfate oxidation, because transcription of the relevant genes would no longer be blocked. A signal-transducing unit would be dispensable in this case. On the other hand, if SoxT1B were responsible for import of oxidizable sulfur, it should be essential for thiosulfate oxidation even when the genes for other components of the sulfur-oxidizing machinery are constitutively expressed.

To differentiate between these possibilities, the transcription of indicator genes was compared by RT-qPCR in the absence versus the presence of thiosulfate. We chose the genes *soxXA* and *shdrA* because they encode central components of thiosulfate oxidation in the periplasm and sulfane sulfur oxidation in the cytoplasm, respectively. The transcription of *soxT1A* and *soxT1B* was also followed. While thiosulfate increases transcript abundance for *soxT1A*, *shdrA* and *soxXA* in the *H. denitrificans* reference strain[8], this is not the case for the strain lacking the *soxT1B* gene (Fig. 5). The thiosulfate oxidation-negative phenotype of this strain is therefore explained by a lack of enzymes required for the degradation of the sulfur substrate. When the gene for the SoxR regulator was deleted together with Δ*soxT1B* from *H. denitrificans* Δ*tsdA*, this resulted in a thiosulfate oxidation-positive phenotype. Transcription was high for *soxT1A, shdrA* and *soxXA* irrespective of the presence of thiosulfate (Fig. 5). The constitutive expression of these genes is caused by the lack of the transcriptional repressor. In the next step, a strain was constructed that lacks genes *soxT1B* and *shdrR*. This strain behaves differently from *H. denitrificans* Δ*tsdA* Δ*soxT1B* Δ*soxR*. It cannot oxidize thiosulfate and the substrate does not induce substantial increase of transcript abundance of the tested sulfur oxidation genes (Fig. 5). SoxR is present in this strain and we suggest that it acts as the major regulator, that prevents transcription even in the presence of thiosulfate when the signal-transducing SoxT1B is not available. This conclusion is particularly true for the bona fide *sox* genes that encode the enzymes that initiate thiosulfate oxidation in the periplasm, i.e., SoxXA, SoxB, and SoxYZ. Note that the *soxXA* transcript abundance in the Δ*tsdA* Δ*soxT1B* Δ*shdrR* strain is low (on the same level as in the reference strain grown in the absence of thiosulfate) and not affected by thiosulfate, as also observed for the Δ*tsdA* Δ*soxT1B* strain. This not only fully explains the inability of the strains to grow on thiosulfate but also indicates that sHdrR plays a subordinate role in the transcription of the *sox* genes.

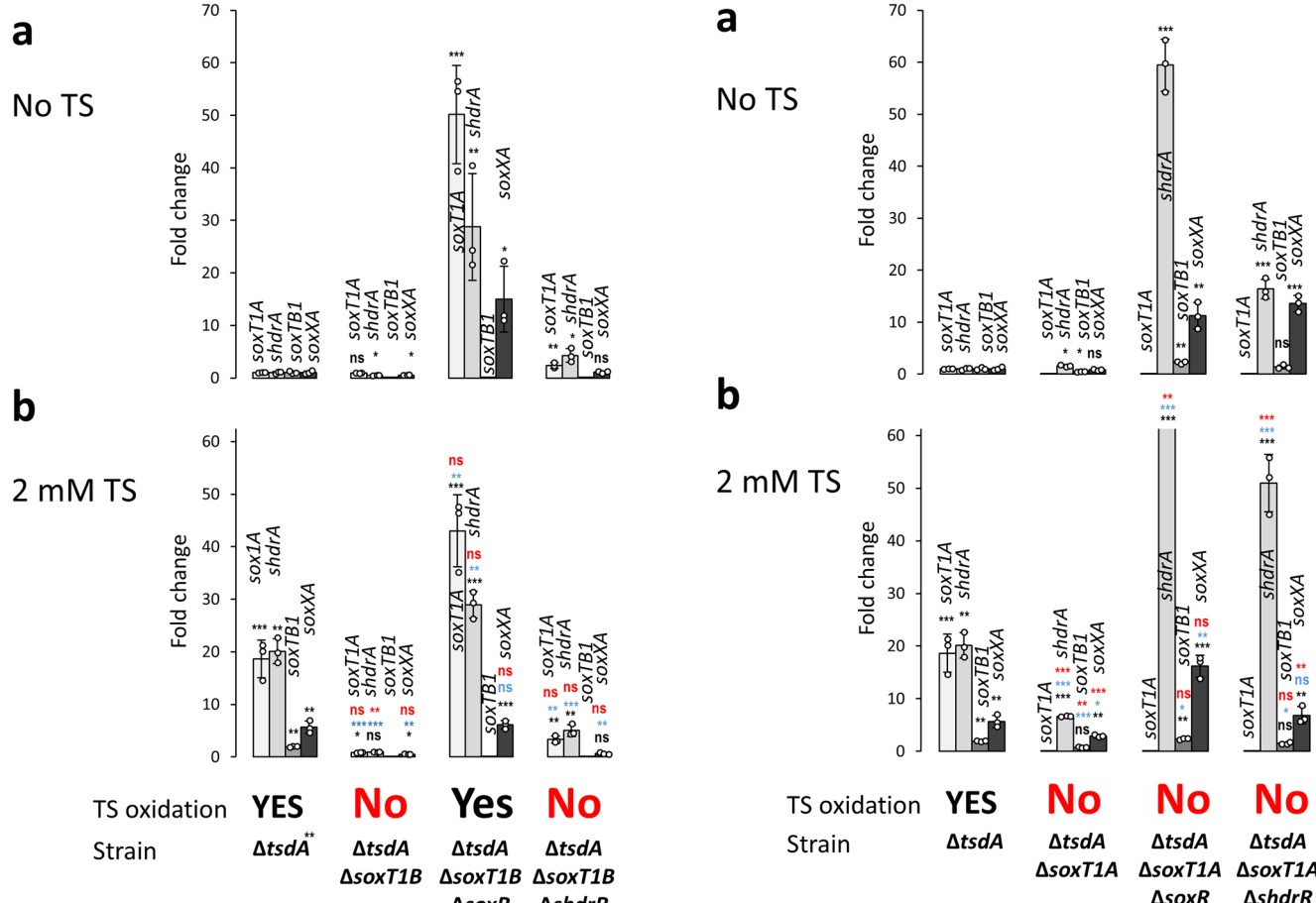

**Fig. 5 | Transcription of sulfur oxidation genes in *H. denitrificans* strains lacking *soxT1B*.** RT-qPCR analysis is shown for four indicative genes in three different *H. denitrificans* Δ*soxT1B* strains in the absence (**a**) and in the presence of 2 mM thiosulfate (**b**). The ability of the strains to oxidize thiosulfate is indicated. The growth experiments are shown in full in Supplementary Fig. 7. All strains grew equally well on methanol in the absence of thiosulfate (Supplementary Fig. 6a). Three parallel experiments were performed to obtain the averages and standard deviation. Data are presented as means ± SD. Individual data points are indicated. One-way ANOVA was performed to calculate *p*-values. Asterisks indicate statistically significant differences (*$p < 0.05$, **$p < 0.01$, ***$p < 0.001$, ns = ($p > 0.05$)). Color coding: black: differences compared to strain Δ*tsdA* in the absence of thiosulfate; blue, differences compared to Δ*tsdA* in the presence of thiosulfate; red, differences compared to the same strain in the absence of thiosulfate. TS thiosulfate, ns not significant.

**Fig. 6 | Transcription of sulfur oxidation genes in *H. denitrificans* strains lacking *soxT1A*.** RT-qPCR analysis for four indicative genes in three different *H. denitrificans* Δ*soxT1A* strains in the absence (**a**) and in the presence of 2 mM thiosulfate (**b**). Note that the fold change for *shdrA* transcript abundance was 201.7 ± 35.5 and thus far exceeds the y-axis range appropriate for all other cases. The ability of the strains to oxidize thiosulfate is indicated. The growth experiments are shown in full in Supplementary Fig. 7. All strains grew equally well on methanol in the absence of thiosulfate (Supplementary Fig. 5b). Three parallel experiments were performed to obtain the averages and standard deviation. Data are presented as means ± SD. Individual data points are indicated. One-way ANOVA was performed to calculate *p*-values. Asterisks indicate statistically significant differences (*$p < 0.05$, **$p < 0.01$, ***$p < 0.001$, ns = ($p > 0.05$)). Color coding: black: differences compared to strain Δ*tsdA* in the absence of thiosulfate; blue, differences compared to Δ*tsdA* in the presence of thiosulfate; red, differences compared to the same strain in the absence of thiosulfate. TS thiosulfate, ns not significant.

In conclusion, the described experiments show that SoxT1B is dispensable for thiosulfate oxidation and that the import of sulfur for further oxidation is not its primary function. Instead, all results are consistent with a signal transduction function.

### Role of SoxT1A in *H. denitrificans*

Like SoxT1B, the related membrane protein SoxT1A could in principle act either as an importer of sulfur for further oxidation in the cytoplasm or as a means of transmitting the information that oxidizable sulfur is available externally. To decide between the two possibilities, the strain *H. denitrificans* Δ*tsdA* Δ*soxT1A* was constructed, phenotypically characterized and studied concerning *shdr* and *sox* gene transcription (Fig. 6). The strain proved to be thiosulfate oxidation negative and accordingly did not show any growth retardation in the presence of thiosulfate (Supplementary Fig. 8), although transcript abundance for *shdrA* and *soxXA* increased significantly in the presence of thiosulfate (Fig. 6).

Further insights were obtained when *H. denitrificans* strains Δ*tsdA* Δ*soxT1A* Δ*soxR* and Δ*tsdA* Δ*soxT1A* Δ*shdrR* were studied. Both strains

show a very high transcript abundance for *soxXA* and *shdrA* in the absence as well as in the presence of thiosulfate. Thus, in principle, sufficient Sox and sHdr proteins are present in SoxT1A-deficient strains to allow thiosulfate oxidation. Still, all SoxT1A-deficient strains are unable to oxidize thiosulfate, suggesting an essential function of SoxT1A in the overall sulfur oxidation pathway.

Remarkably, transcript level increases for *shdrA* and *soxXA* in *H. denitrificans* Δ*tsdA* Δ*soxT1A* caused by the presence of thiosulfate appear low, 6.62 ± 0.13 and 2.92 ± 0.36 fold, respectively, when compared to the *soxT1A*-deficient strains that additionally lack either one of the studied regulator genes (Fig. 6). It is important to note that the increases determined for the Δ*tsdA* Δ*soxT1A* strain are in a similar range though not exactly the same as observed for the Δ*tsdA* reference strain by RT-qPCR (20.11 ± 2.39 fold and 5.68 ± 1.18 fold for *shdrA* and *soxXA*, respectively, Fig. 3c) and in our mRNA-Seq experiments (19.3 and 6.7–8.9 fold, Supplementary Table 2). Increasing transcript abundance in this moderate range allows

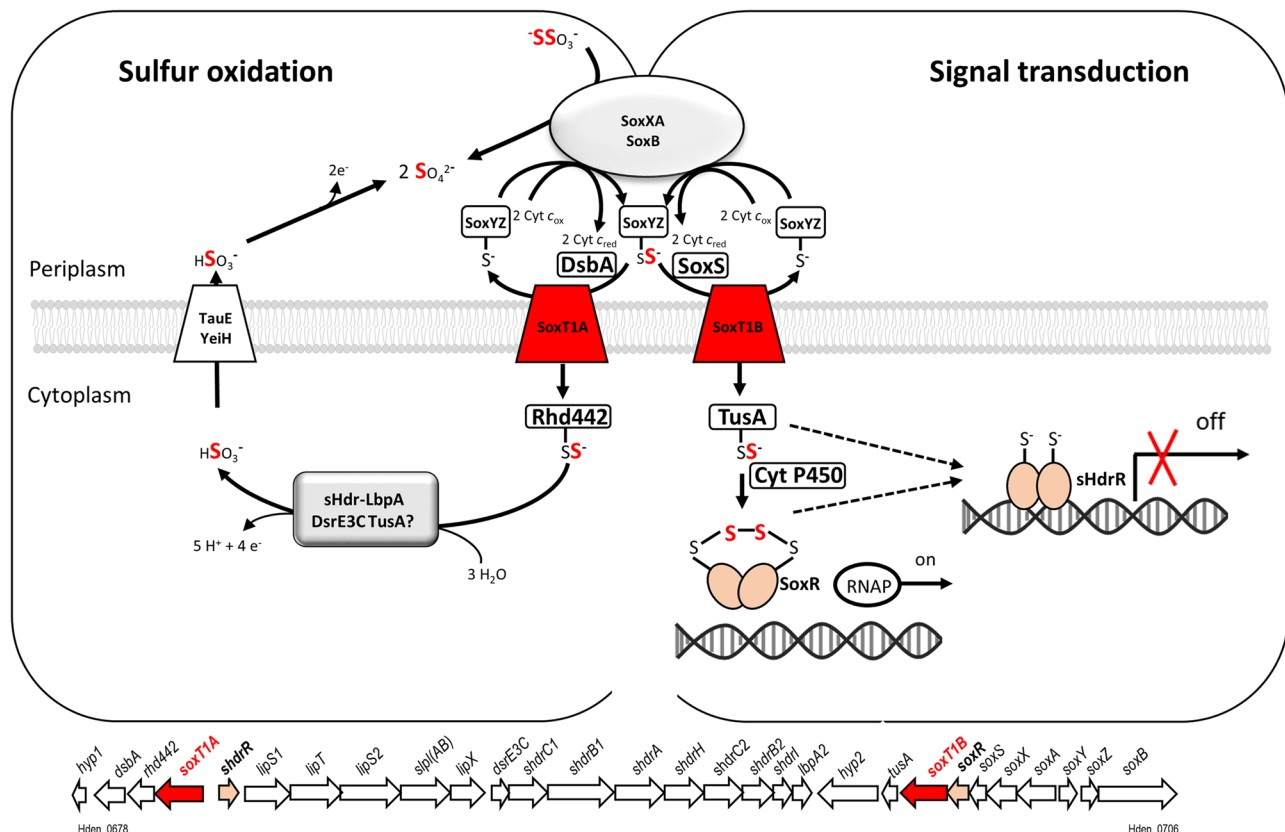

**Fig. 7 | Model of thiosulfate oxidation in *H.denitrificans* integrating the sulfur transport and signal transduction functions of SoxT1A and SoxT1B, respectively.** TauE, putative sulfite exporter encoded by Hden_0720[28]. Transcripts are 2.6-fold more abundant when thiosulfate is present (Supplementary Table 1). YeiH (Hden_0834) is another candidate for sulfite export, with increased transcript abundance in the presence of oxidizable sulfur. The *lipS1,lipT, lipS2, slpl(AB)* and *lipX* genes encode enzymes that assemble the cofactor on the lipoate-binding protein LbpA2[30]. RNAP, RNA polymerase. Sulfur atoms printed in red stem from the sulfane sulfur atom of thiosulfate, the oxidation of which is initiated in the periplasm. sHdrH and sHdrI are soluble, cytoplasmic proteins of unknown function. *hyp1*, encodes a56-aa transmembrane protein; *hyp2* encodes a putative cytochrome P450.

thiosulfate oxidation. However, if one of the repressors is missing, the whole system apparently loses its balance resulting in extremely high transcription levels (Fig. 6) and thus energy-intensive biosynthesis of the sulfur-oxidizing system including a complete assembly machinery for its specific lipoate-binding proteins. Further studies are needed to elucidate how exactly the fine-tuning of transcription rates to metabolic needs is achieved in the reference strain, whether SoxT1A has a share in the signaling process, even if sulfur import appears to be its major function, and whether transcriptional regulators other than SoxR and sHdrR are involved in this process. The latter seems likely since the transcription of genes for LuxR, LysR, and Fur-type regulators is significantly affected by the presence of thiosulfate (Supplementary Table 2 and Supplementary Fig. 3).

## Discussion

Here, we provide information on the distribution and phylogeny of YeeE-like transporters in sulfur-oxidizing prokaryotes and even more importantly, we assign fundamentally different functions to two of these proteins, SoxT1A and SoxT1B, that co-occur in the same Alphaproteobacterium, *H. denitrificans*. The completely different regulation of the respective genes upon exposure of the organism to thiosulfate is the first milestone for functional assignment. Expression of *soxT1A* is highly increased, while *soxT1B* expression is hardly affected at all by the presence of the reduced sulfur compound. All of our observations are consistent with the central role of SoxT1A in sulfur oxidation. The amount of SoxT1A molecules in the cells is increased to ensure efficient import of sulfur into the cytoplasm where it is further processed by the sHdr-LbpA system (Fig. 7). To the best of our knowledge, *H. denitrificans* SoxT1A is the only experimentally

demonstrated sulfur importer in dissimilatory sulfur-oxidizing prokaryotes. However, it does not provide a general solution because it not even occurs in all sulfur oxidizers using the cytoplasmic sHdr pathway. SoxT transporters are completely absent genomes with type II sHdr, even though some of these organisms harbor the capacity for Sox-driven thiosulfate oxidation (Fig. 2).

SoxT1B functions as a signal transducing module. The same function can be assumed for the SoxT proteins in Alphaproteobacteria with complete Sox systems. In these organisms, thiosulfate is completely oxidized to sulfate in the periplasm and accordingly they lack cytoplasmic sulfur-oxidizing enzymes. As a consequence, there is no need for mass import of sulfur as carried out by SoxT1A. In full agreement with these conclusions, a function of SoxT from *Pseudaminobacter salicylatoxidans* in the transport of an inducer to the cytosol to activate the transcriptional regulator SoxR has been suggested[37].

The genetic neighborhood of the *soxT1A* and *soxT1B* genes provides a basis for a model of how sulfur might be presented to the transporters, transported through them, and delivered to their final targets (Fig. 7). In immediate vicinity to and in the same direction of transcription with *soxT1A*, a gene (Hden_0679) is located that encodes a periplasmic DsbA-like thioredoxin with two thioredoxin-like cysteine motifs (Cys-$X_2$-Cys), one of which resides at the very carboxy-terminal end of the protein. Thioredoxins serve as general protein disulfide oxidoreductases that interact with a broad range of proteins by a redox mechanism based on reversible oxidation of two cysteine thiol groups to a disulfide, accompanied by the transfer of two electrons and two protons (IPR013766). We consider the possibility that the *H. denitrificans* DsbA is involved in the release of sulfane sulfur from the persulfidated periplasmic sulfur carrier SoxYZ and

that the sulfur is then transferred into the cytoplasm through SoxT1A. In the cytoplasm, the sulfur is further handled by cytoplasmic Rhd442 (Hden_680), a protein that we recently characterized as a rhodanese-like sulfur transferase[6]. From there, the sulfur is delivered to the sulfur transferase DsrE3C and finally oxidized to sulfite by the sHdr-LbpA system, possibly involving TusA[5,6]. Hden_0678 encodes short 56 aa membrane protein, lacking cysteine residues and consisting of one central transmembrane helix (aa 12 to 27) with the N-terminus predicted to reside in the cytoplasm. Functional assignment is currently not possible.

The genes in the vicinity of *soxT1B* appear to encode a second module dedicated to the transport of sulfur, albeit for a different purpose. As suggested earlier, it is conceivable that sulfur bound to the sulfur carrier protein SoxYZ is in this case presented to the transporter by the periplasmic thiol–disulfide oxidoreductase SoxS[38]. In fact, SoxS from *P. denitrificans* specifically binds SoxY[38]. Once in the cytoplasm, the sulfur transferase TusA[6,18] is a likely acceptor protein for the sulfur. This idea is corroborated by recent findings for the *E. coli* thiosulfate transporter TsuA[14,16]. TsuA belongs to same family as the SoxT transporters and the TusA-like TsuB protein was shown to be essential for TsuA mediated thiosulfate uptake in vivo. TsuB can cleave thiosulfate resulting in persulfidation of its conserved cysteine and the release of sulfite. In *H. denitrificans*, sulfur atoms could be passed on from TusA to either one or both of the transcriptional repressors encoded in the *shr-lbpA-sox* genomic region. For SoxR, we showed that it forms an intramolecular sulfur bridge between two conserved cysteines[8]. The formation of this bridge is the trigger to detach from its target DNA and thus to enable transcription. We assume that sHdrR, which closely resembles SoxR[8], functions accordingly. Whether SoxR and/or sHdrR are indeed loaded with sulfur in a reaction mediated by TusA or rather directly by the sulfur species transported through SoxT1B, cannot be answered on the current data basis.

We believe that it is advantageous for *H. denitrificans* to have the genetic potential for two different SoxT1 modules, one dedicated to signal transduction and one dedicated to import sulfur for further oxidation. Only a few SoxT1B protein molecules have to be present to serve their purpose in a signaling cascade. On the other hand, cells probably have to be richly equipped with SoxT1A molecules that have to ensure mass transport of sulfur as part of energy conservation. Separating the two functions has the advantage that large quantities of SoxT1A are only synthesized when they are really needed.

The exact chemical nature of the sulfur species transported by SoxT1A and SoxT1B requires further investigation, although both transporters are structurally similar to the characterized *S. thermophila* thiosulfate transporter TsuA (YeeE) and share the substrate binding pocket (Supplementary Fig. 9). The substrate for TsuA (YeeE) is thiosulfate. At present, we cannot rule out the possibility that in *H. denitrificans* a small fraction of the thiosulfate available for oxidation is itself used as a signal molecule, channeled through SoxT1B and then cleaved by TusA, as proposed for TsuA. In mutant strains lacking SoxT1A, sulfane sulfur oxidation in the cytoplasm is halted because there is no supply of substrate, thus causing inability to oxidize thiosulfate altogether. Notably, in Δ*soxT1A* strains higher transcription is observed for *shdrA* and *soxXA* in the presence than in the absence of thiosulfate (Fig. 6), pointing at thiosulfate itself as the signal molecule. However, other results presented in Fig. 6 are not fully in line with this possibility: Transcript levels of *shdrA* and *soxXA* in the *H. denitrificans* Δ*soxT1A* strain in the presence of thiosulfate are lower than those in the Δ*soxT1A* Δ*soxR* strain, although the signal transduction pathway is intact in that mutant. If the signaling molecule transported by SoxT1B were thiosulfate, full release of transcriptional repression would be expected. On the other hand, unknown layers of regulation may be involved in the fine tuning of *shdr* and *sox* gene transcription and it appears premature to draw conclusions about the transported compounds based on the results thus far available.

Concerning SoxT1A, thiosulfate can be excluded as its substrate. In *H. denitrificans*, thiosulfate degradation is initiated in the periplasm, where it is attacked by the periplasmic proteins of the truncated Sox system. SoxXA

and SoxB release sulfate and SoxYZ-bound sulfane sulfur, which then has to be further processed in the cytoplasm. Thiosulfate oxidation is not possible in the absence of SoxXA and SoxYZ[7], proving that thiosulfate cannot be taken up and processed in the cytoplasm.

In fact, a transport by passing sulfur from the SoxY cysteine along the three cysteines lining the central channel of the SoxT1 proteins is conceivable. Ikei and coworkers suggest that interaction of thiosulfate with the cysteine residues occurs via S—H—S hydrogen bonds[16]. The three cysteine residues in TsuA (YeeE) are linearly located at intervals of ~7 Å, while disulfide bonds are usually about 2.05 Å in length, and 3.0 Å is taken as the cutoff for disulfides in the PDB database. It is therefore questionable whether sulfur atoms can be directly transferred from one cysteine sulfur to the next. Free HS⁻ ions or short polysulfides ($^-S$-$S_n$-$S^-$), that are possibly formed by the action of the periplasmic protein disulfide oxidoreductases DsbA and SoxS, are alternatives and conceivable substrates for cytoplasmic sulfur transferases such as Rhd442 or TusA. Even a direct reaction of polysulfides with the transcriptional repressors, as occurs in vitro[8], is conceivable.

## Methods

### Bacterial strains, plasmids, primers, and growth conditions

Supplementary Table 3 lists the bacterial strains, and plasmids that were used for this study. *Escherichia coli* strains were grown on complex lysogeny broth (LB) medium[39]. *E. coli* 10β was used for molecular cloning. *H. denitrificans* strains were cultured in minimal medium kept at pH 7.2 with 100 mM 3-(*N*-Morpholino)propanesulfonic acid (MOPS) buffer as previously described[28]. Media contained 24.4 mM methanol. Antibiotics for *E. coli* and *H. denitrificans* were used at the following concentrations (in µg ml⁻¹): ampicillin, 100; kanamycin, 50; streptomycin, 200; chloramphenicol, 25.

### Recombinant DNA techniques

Standard techniques for DNA manipulation and cloning were used unless otherwise indicated[40]. Restriction enzymes, T4 ligase and Q5 polymerase were obtained from New England Biolabs (Ipswich, UK) and used according to the manufacturer's instructions. Oligonucleotides were obtained from Eurofins Genomics Germany GmbH (Ebersberg, Germany). Plasmid DNA from *E. coli* was purified using the GenJET Plasmid Miniprep kit (Thermo Scientific, Waltham, USA). Chromosomal DNA from *H. denitrificans* strains was prepared using the Simplex Easy DNA Extract Kit (GEN-IAL GmbH, Troisdorf, Germany). DNA fragments were extracted from agarose gels using the GeneJET Gel Extraction Kit (Thermo Scientific, Waltham, USA).

### Construction of *H. denitrificans* mutant strains

Plasmids for reverse genetics in *H. denitrificans* were constructed using the suicide plasmid pk18*mobsacB*[41] and the tetracycline cassette from pHP45Ω-Tc[42] on the basis of previously published procedures[28,29]. For markerless *in frame* deletion of the individual *H. denitrificans* *soxT1A* and *soxT1B* genes by splicing overlap extension (SOE)[43], PCR fragments were constructed using the primers listed in Supplementary Table 3. The *soxT1A* or *soxT1B* fragments were inserted into pk18*mobsacB* using XbaI and SaII or XbaI and PstI restriction sites, respectively. The SmaI-excised tetracycline cassette from pHP45Ω-Tc[42] was inserted into the SmaI site, resulting in plasmids pK18*mobsacB*-Δ*soxT1A*-Tc and pK18*mobsacB*-Δ*soxT1B*-Tc. Another plasmid was constructed for concomitant deletion of *soxR* and *soxT1B* by SOE PCR with primers P1 fwd up hden_0700, P5 fwd down hden_soxR/soxT1B, P6 rev down hden_ soxR/soxT1B and P7 rev up hden_ soxR/soxT1B (Supplementary Table 3). The PCR fragment was cloned into the XbaI and PstI sites of pk18*mobsacB*-Tc[7]. For chromosomal complementation of the *H. denitrificans* Δ*tsdA* Δ*soxT1B* strain, the *soxT1B* gene was amplified together with upstream and downstream regions using primers SoxT1B_Del_Up_Fw and SoxT1B_Del_Down_Rev and cloned into the XbaI/PstI sites of pk18*mobsacB*-Tc. For chromosomal integration of the genes encoding SoxT1B Cys24Ser, SoxT1B Cys98Ser and SoxT1B Cys304Ser, the modified genes and upstream and downstream sequences were

amplified by SOE PCR using the appropriate primers listed in Supplementary Table 3.

All final constructs were electroporated into the desired *H. denitrificans* strains and transformants were selected using previously published procedures[28,29]. *H. denitrificans ΔtsdA* served as acceptor for plasmids pK18*mobsacB*-*ΔsoxT1B*-Tc, pK18*mobsacB*-*ΔsoxT1A*-Tc and pk18*mobsacB*_Tc_*ΔsoxR/soxT1B*. *H. denitrificans ΔtsdA ΔshdrR* and *H. denitrificans ΔtsdA ΔsoxR* served as strain backgrounds for deletion of *soxT1A*. The *soxT1B* deletion was also established in the *H. denitrificans ΔtsdA ΔshdrR* strain. The plasmids for complementation and cysteine exchanges of SoxT1B were transferred into *H. denitrificans ΔtsdA ΔsoxT1B* in all cases, single crossover recombinants were Cm$^r$ and Tc$^r$. Double crossover recombinants were Tc$^s$ and survived in the presence of sucrose due to loss of both, the vector-encoded levansucrase (SacB) and the tetracyclin resistance gene. The genotype of the *H. denitrificans* strains generated in this study was confirmed by PCR.

### Characterization of phenotypes, quantification of sulfur compounds, and biomass content

Growth experiments with *H. denitrificans* were run in medium with 24.4 mM methanol in Erlenmeyer flasks or 96-well microtiter plates as described earlier[7]. 2 mM thiosulfate was added when needed. Biomass content, thiosulfate, and sulfite concentrations were determined by previously described methods[7,44]. All growth experiments were repeated three to five times. Representative experiments with two biological replicates for each strain are shown. All quantifications are based on at least three technical replicates.

### Expression studies based on RT-qPCR

Total RNA of the relevant *H. denitrificans* strains was isolated from cells harvested in mid-log phase according to an established procedure[8]. RNA samples of 100 ng were used for RT-qPCR analysis which was performed with the primers listed in Supplementary Table 3 following the method described in ref. [8].

### Genome-wide transcriptomic analysis of *H. denitrificans ΔtsdA* in the absence and presence of thiosulfate

For transcriptome sequencing (RNA-Seq), *H. denitrificans ΔtsdA* was cultured in 50 ml minimal medium containing either 24.4 mM methanol or 24.4 mM methanol plus 2 mM thiosulfate in 200 ml Erlenmeyer flasks at 30 °C with shaking at 200 rpm to early log phase. Cells from 20 ml culture were harvested and flash frozen in liquid N$_2$ and stored at −70 °C. From the frozen pellets, the RNA was purified with the FastGene RNA Premium Kit (NIPPON Genetics EUROPE, Düren, Germany) according to the manufacturer's instructions. A modification was introduced regarding the cell lysis step. After the addition of the lysis buffer that contained 1% (v/v) 2-mercaptoethanol, cells were disrupted by bead beating (Bead Ruptor 12 Bead Mill Homogenizer, Omni International, Kennesaw, GA, USA) for three cycles of 30 s at maximum speed and incubation on ice for 1 min. RNA quality was checked on 1% agarose gels and its concentration was measured using NanoPhotometer NP80 (IMPLEN, Munich, Germany). The RNA was shipped on dry ice to Eurofins Genomics GmbH (Ebersberg, Germany). The subsequent analysis pipeline included rRNA depletion, library preparation (mRNA fragmentation, strand-specific cDNA synthesis), Illumina paired-end sequencing (2 ×150 bp, minimum 10 MB reads)), and bioinformatic analysis (mapping against the reference genome, identification and quantification of transcripts, pairwise comparison of expression levels and determination of significant fold differences) and was conducted by the company.

### Generation of datasets for phylogenetic analyses

Archaeal and bacterial genomes were downloaded from Genome Taxonomy Database (GTDB, release R207). In GTDB, all genomes are sorted according to validly published taxonomies, they are pre-validated and have high quality (completeness minus 5*contamination must be higher than 50%). One representative of each of the current 65,703 species clusters was analyzed. Open reading frames were determined using Prodigal[45] and subsequently annotated for sulfur-related proteins via HMSS2[34]. Annotation was extended by HMMs from TIGRFAMs[46] and Pfam[47] databases representing the 16 syntenic ribosomal proteins RpL2, 3, 4, 5, 6, 14, 15, 16, 18, 22, and 24, and RpS3, 8, 10, 17, and 19. A type I sHdr system was considered to be present if the core genes *shdrC1B1AHC2B2* were present in a syntenic gene cluster. For a type II sHdr system gene cluster *shdrC1B1AHB3* and *etfAB* had to be present in a single syntenic gene cluster[29,48].

### Phylogenetic tree inference

For species tree inference, results for each ribosomal protein were individually aligned, trimmed and subsequently concatenated before they were used for phylogenetic tree construction. Proteins were aligned using MAFFT[49] and trimmed with BMGE[50] (entropy threshold = 0.95, minimum length = 1, matrix = BLOSUM30). Alignments were then used for maximum likelihood phylogeny inference using IQ-TREE v1.6.12[51] implemented on the "bonna" high-performance clusters of the University of Bonn. The best-fitting model of sequence evolution was selected using ModelFinder[52]. Branch support was then calculated by SH-aLRT (2000 replicates)[53], aBayes (2000 replicates)[54], and ultrafast bootstrap (2000 replicates)[55]. Finally, trees were displayed using iTol[56].

### Statistics and reproducibility

Experimental data are expressed as the mean ± standard deviation of the mean of the number of tests stated for each experiment. All analysis was reproduced in at least three independent experiments. One-way ANOVA was performed to calculate *p*-values, with values < 0.05 indicating statistical significance.

### Reporting summary

Further information on research design is available in the Nature Portfolio Reporting Summary linked to this article.

### Data availability

RNA-Seq raw data files and processed data files are available via the NCBI GEO repository (accession number GSE278992). Protein accession numbers and species names for Fig. 1 are available at https://github.com/WandaFlegler/Masterarbeit/blob/main/Galaxy3-%5BBMGE_Cleaned_sequences_Fasta%5D.fasta.treefile. The authors declare that all other data supporting the findings of this study are available within the article (and its supplementary information files). The source data underlying Fig. 2 are provided as a source data file (Supplementary data 1). The source data underlying Figs. 3, 4, 5, 6 and Supplementary Figs. 6, 7, and 8 are provided as a source data file (Supplementary data 2).

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

## Acknowledgements

This work was funded by the Deutsche Forschungsgemeinschaft (Grants Da351/8-2, Da 351/13-1, and Da 351/14-1). J.L. was financed by a Scholarship from the Chinese Scholarship Council and T.S.T. received a scholarship from the Studienstiftung des Deutschen Volkes. We thank Stefania de Benedetti for help with RNA isolation and Marc Gregor Mohr for help with handling RNAseq raw data.

## Author contributions

Conceptualization, J.L. and C.D.; investigation, J.L., F.G., H.Y.H., J.N.K., N.H., W.A.F., and T.S.T.; validation, J.L., T.S.T., and C.D.; writing-original draft preparation, J.L. and C.D.; writing-review and editing, J.L. and C.D., visualization, J.L., T.S.T., and C.D.; supervision, C.D.; project administration, C.D.; funding acquisition, C.D. All authors have read and agreed to the published version of the manuscript.

## Funding

## Competing interests

The authors declare no competing interests.
