## [Transparent Peer Review file · Communications Biology]

YeeE-like bacterial SoxT proteins mediate sulfur import for oxidation and signal transduction

Corresponding Author: Dr Christiane Dahl

Version 0:

Reviewer comments:

Reviewer #1

(Remarks to the Author)

The authors of this manuscript previously determined that Sox-TsdA proteins and Hdr-LbpA-Dsr proteins contribute periplasmic and cytoplasmic sulfur compounds metabolism in this bacteria, respectively. Moreover, they had already found differences in the transcriptional mechanisms undergone by soxT1A and soxT1B. In this report, they extended their studies to the effects of SoxT1A and SoxT1B transporters on sulfur metabolism and regulatory systems.

This paper clearly show the impact of sulfur transport on sulfur metabolism and its regulation, as well as the phylogenetic position of the targeted transporters. It is quite interesting that two similar transporters are independently involved in sulfur metabolism and signal regulation, respectively. This study is important not only for the regulatory mechanisms of sulfur metabolism, but also for examining the function of transporters in other metabolites. Therefore, I recommend that this paper deserves to be published in Communications Biology after minor revision.

1. I think that the *tsdA* mutant cannot metabolize thiosulfate to tetrathionate; however, I cannot completely understand why you used this mutant strain as a reference. Explaining the use of this mutant strain in the study support the reader's understanding.
2. The explanations of Fig. 4A and B in the text are reversed; therefore, either the explanations or the A, B in the figure need to be swapped.
3. The authors need to describe the phenotype of the *soxT1A* mutant near line 322, citing Figure S6.

Reviewer #2

(Remarks to the Author)

In this study, the authors identified two YeeE/YedE-family proteins occurred in the neighborhood of sulfur-metabolizing cluster in *Hyphomicrobium denitrificans*. Phylogenetic analysis assigned these two transmembrane proteins into SoxT1 group, term SoxT1A and SoxT1B. By measuring the transcript levels of marker genes and TS oxidation in the absence or presence of TS in Δ soxT1A or Δ soxT1B in combination with Δ soxR or Δ shdrR, the authors claimed that SoxT1A function as sulfur importer and SoxT1B as signal transducer. This is the central statement of this paper, however, there are many issues to be addressed.

Major comments:

1. Why use the Δ tsdA as reference strain? The background knowledge of Δ tsdA is needed.
2. 'The transcript abundance for soxT1A increased more than tenfold upon addition of thiosulfate in the *H. denitrificans* Δ tsdA reference strain, while soxT1B expression remained essentially unaffected'. There is no figure or table for this conclusion.
3. 'A similar role may be played by YeiH in *H. denitrificans*'. The protein sequence similarity of YeiH should be provided to support this conclusion.
4. The fold change expression of mRNA-Seq should be consistent, either log2 or not, for all described genes.
5. '...the transcription of only three genes in the genomic sulfur oxidation region shown in Fig. 3a proved unaffected in the mRNA-Seq experiment...'. If so, the column of *shrR*, *soxT1B* and *soxR* in the presence of thiosulfate should be comparable to that in the absence of thiosulfate. In this figure, the fold change of these three genes in the presence of thiosulfate is near zero, meaning that they are far downregulated, rather than unaffected by thiosulfate. Please explain it.
6. Fig. 3b, the transcript levels of *soxR* and *shrR* in the presence of thiosulfate are needed to support the conclusion that '...

soxT1B expression, along with that of soxR and also shdrR, does not change significantly...’.

7. Fig. 3b, the transcript levels of tested genes in the absence of thiosulfate in Δ tsdA Δ soxR and Δ tsdA Δ shdrR should be provided to draw this conclusion that ‘In the repressor-negative strains, soxT1A expression is high even in the absence of thiosulfate, while soxT1B transcript abundance is hardly affected’.

8. Fig. 3b, why are the transcript levels of all tested genes in Δ tsdA Δ shdrR decreased compared to the reference strain Δ tsdA?

9. Fig. 4a and Fig. S5b, assuming that soxT1B function as signal transducer, the TS concentration should be decreased since the reaction of TS in the periplasm and the import system are intact. Please explain it.

10. How to exclude the possibility that membrane protein serves as toxic product exporter? Please explain it.

11. Fig. 5 and Fig. 6, the fold changes of tested genes in the reference strain should be present for comparison.

12. Fig. 6. assuming SoxT1A import certain sulfur compound derived from periplasmic TS oxidation, why is the transcript levels of shdrA and soxXA in Δ soxT1A far lower than that in Δ soxT1A Δ soxR by TS addition, since the signal transducing pathway is intact? Why is the derepress effect of the knockout of shdrR on shdrA and soxXA much more significant in Δ soxT1A Δ shdrR compared to that Δ soxT1B Δ shdrR (Fig. 5)? Why is the fold change value of shdrA so dramatically high in Δ soxT1B Δ soxR in the presence of TS, unlike that in Δ soxT1A Δ soxR in the same condition?

13. Fig. S6b, why did the TS concentration not decrease in Δ tsdA Δ soxT1A if the substrate transported by SoxT1A is certainly not TS since effective oxidation of TS occurred in periplasm (in discussion part)?

Minor comments:

1. ‘...Serratia sp. ATCC39006...’ in the Introduction part, ‘sp.’ should be regular

2. ‘Members of the YeeE/YeeD family...’ should be ‘Members of the YeeE/YedE family’

3. ‘mRNA transcripts of 47 genes showed lower mRNA abundance than in its presence’ should be ‘mRNA transcripts of 47 genes showed lower mRNA abundance than in its presence’

4. ‘A cisimilar role may be played by YeiH ...’ should be ‘A similar role may be played by YeiH...’

5. The positions of Fig. 4a and 4b should be switched.

6. The y-axis of Fig. 6 should be changed into two-segments to show the value of shdrA.

Reviewer #3

(Remarks to the Author)

Li et al. have investigated the physical functions of SoxT1A and SoxT1B in *Hyphomicrobium denitrificans*. They observed some features that potentially explain how sulfur import into the cytoplasm and how regulators sense external sulfur, but the conclusion remains somewhat speculative. The work was not enough to prove the conclusion that SoxT1A delivers sulfur to the cytoplasm for its further oxidation and SoxT1B serves as a signal transduction unit for the transcriptional repressor SoxR. Below are concerns that should be addressed.

1. *H. denitrificans* contains many thiosulfate oxidation pathways. The authors mentioned that further thiosulfate oxidation steps take place in the cytoplasm, which requires the import of sulfane sulfur substrate and export of sulfite. The product of further oxidation in the cytoplasm is sulfite, which is also toxic to cell. Why *H. denitrificans* chooses this risky pathway to oxidize thiosulfate rather than directly oxidizing thiosulfate in the periplasm by TsdA or Sox system. I would recommend the authors clarify the using orders of these thiosulfate oxidation pathways in *H. denitrificans*. Whether the thiosulfate oxidation pathway in the cytoplasm is useless when thiosulfate could be oxidized in the periplasm by TsdA.

2. In Fig. 4, why the authors use Δ tsdA as the control rather than using WT strain? The growth curves and thiosulfate consumption rate of wild-type strain should be shown. Little was known about the differences of thiosulfate oxidation ability between WT and Δ tsdA strain.

3. The author considered that Δ soxT1B may act as a signal transduction protein. However, the evidences for supporting this point are insufficient. The signal molecular transformed by membrane protein SoxT1B was failed to be identified. Why the sulfur oxidation substrates imported into the cytoplasm fail to serve as a sensing signal molecular for SoxR? Why a specific transmembrane protein was applied only for signal transducing but not for sulfur oxidation substrate transforming. To confirm the function of SoxT1B, it is suggested that figure out which kind of sulfur compound serves as the signal of SoxR. Purify SoxT1B and its mutant proteins and detect the affinity changes between the signal molecular and the proteins. Conduct molecular docking to reveal the mechanism how SoxT1B transfers this signal molecular and then figure out the signal binding pocket in SoxT1B.

4. SoxT1A mutants are sulfur oxidation-negative despite high transcription levels of sulfur oxidation genes. The authors mentioned that SoxT1A delivers sulfur to the cytoplasm for its further oxidation. The only data to support this point comes from RT-qPCR analysis. To further confirm this conclusion, protein expression level of SoxXA and ShdrA sulfur oxidation enzymes should be detected to exclude the post-transcriptional regulation in sulfur oxidation. Detect whether the phenotype of Δ soxT1A mutant could be complemented by the genes responsible for thiosulfate import in other Alphaproteobacteria. Detect the affinity between SoxT1A and its transport substrate by isothermal titration calorimetry. Construct conserved residue mutants of SoxT1A and detect their ability of thiosulfate oxidation.

5. Page 9 line 250: as shown in Fig. 3b, --soxXA was selected as one of the four indicative genes-- According to the locus diagram, soxX and soxA are two genes and their expression levels were not the same as shown in Fig. 3a. soxXA labeled in the column diagram could be corrected as soxX or soxA.

6. Page 9 line 253-254: In Fig. 3b legend ---the absence of thiosulfate are also shown. All changes are compared to H.

denitrificans Δ tsdA in the absence of thiosulfate----, however, the corresponding results were missing in Fig. 3b.

7. In Fig.5b, Δ soxT1B, Δ soxT1BsoxR and Δ soxT1BshdrR should be corrected as Δ tsdA Δ soxT1B, Δ tsdA Δ soxT1BsoxR and Δ tsdA Δ soxT1BshdrR according to the description in Page10 line281.

8. In Fig.6b, Δ soxT1A, Δ soxT1AsoxR and Δ soxT1AshdrR should be corrected as Δ tsdA Δ soxT1A, Δ tsdA Δ soxT1AsoxR and Δ tsdA Δ soxT1AshdrR according to the description in Page12 line320.

9. In supply information Fig. 4b, Δ tsdA Δ soxT1B Δ shdrR should be corrected as Δ tsdA Δ soxT1A Δ shdrR.

Version 1:

Reviewer comments:

Reviewer #2

(Remarks to the Author)

This manuscript has been improved compared to the previous version. However, some issues are still not be addressed. Here are the specific comments.

1. Fig. 3b, Indications should be added in the RT-qPCR figure showing that the column of Δ tsdA reference strain is in the presence of thiosulfate and the columns of repressor absence strains are in the absence of thiosulfate.

2. Is soxYZ also regulated by SoxR?

3. Fig. 5 and Fig. 6. RT-qPCR of the reference strain is needed in each set of experiment since the absolute abundances of tested genes vary in different experiments. All the changes in Fig. 5 and Fig. 6 have to be calculated based on reference strain of the same set. Even though the response of the reference strain is similar to Fig. 3, the present of reference strain in each set has to be present to show the role of knocked out genes.

4. Fig. 6 The fold change of shdrA (~6-fold) is much lower in the presence of thiosulfate in Δ tsdA Δ soxT1A than that in Δ tsdA (Fig. 3b, ~20 fold). They are definitely not in the same range. It did not make sense if the SoxT1A functions only in certain intermediate into cytoplasm. Also, since SoxR is the major regulator directly control the transcription of sox genes, the absence of soxR should derepress the shdrA to the similar levels in both Δ tsdA Δ soxT1A Δ soxR and Δ tsdA Δ soxT1B Δ soxR. It is odd that the fold change of shdrA in Δ tsdA Δ soxT1A Δ soxR in the presence of thiosulfate is so high (>200 fold). Since the author could not provide more solid evidence to prove the transporter function of SoxT1A, it is not suitable to assert that SoxT1A as sulfur importer especially in the title.

Reviewer #3

(Remarks to the Author)

This study focuses on investigating the function of SoxT1A and SoxT1B involved in sulfur metabolism and signal regulation in H. denitrificans. Although the authors emphasized that their study is novel, the methods for genetic manipulation of H. denitrificans are time-consuming and they are currently not in the position to perform some type of experiments, I still recommend that the experimental data as suggested in the review comments should be obtained before publishing this paper in Communications Biology.

1. The author mentioned that in the presence of thiosulfate, regulator negative strains including Δ tsdA Δ soxR and Δ tsdA Δ shdrR show very little growth during thiosulfate oxidation due to constitutive expression of all sox and shdr genes, and it is virtually impossible to collect enough cell for mRNA extraction during this growth phase. However, in Δ tsdA Δ soxT1B Δ soxR, sox and shdr genes are also constitutive expression, and this strain also has thiosulfate oxidation ability, why its RT-qPCR analyses could be performed. I think these results are confused.

2. The authors considered that H.denitrificans SoxT1A is the only experimentally demonstrated sulfur importer in dissimilatory sulfur-oxidizing prokaryotes. However, which kinds of sulfur compound serving as the transport substrate transformed by SoxT1A failed to be identified in the manuscript. Whether these sulfur compounds could upregulate the expression of SoxT1A should be addressed. The proteins belonging to YeeE/YedE family are responsible for thiosulfate import was confirmed in E.coli previously, the findings that this family proteins have the similar function in H.denitrificans are not surprise. Although this work is interesting, breadth of discovery by the current experimental data is unlikely to result in major advance. Some efforts on the model of how sulfur transported through the transporters should be added in this manuscript. Conducting some biochemical experiments about the interaction between the sulfur molecular and the transporters contribute to the understanding of functions of SoxT1A and SoxT1B in H.denitrificans.

Version 2:

Reviewer comments:

Reviewer #2

(Remarks to the Author)

All my concerns have been addressed. I think the revised manuscript is ready for acceptance.

Reviewer #3

(Remarks to the Author)

This manuscript has been significantly improved compared to the previous version. I recommend that this manuscript deserves to be published in Communications Biology.

Manuscript ID.: **COMMSBIO-24-2898**

Title: "Yee-like bacterial SoxT proteins mediate sulfur import for oxidation and signal transduction"

Corresponding Author: Christiane Dahl

All Authors: Jingjing Li, Fabienne Göbel, Hsun Yun Hsu, Julian Nikolaus Koch, Natalie Hager, Wanda Antonia Flegler, Tomohisa Sebastian Tanabe, Christiane Dahl

We thank the reviewers for the helpful comments. Please find our point by point answers below. For better readability our answers are printed in italics.

Reviewers' comments:

Reviewer #1

The authors of this manuscript previously determined that Sox-TsdA proteins and Hdr-LbpA-Dsr proteins contribute periplasmic and cytoplasmic sulfur compounds metabolism in this bacteria, respectively. Moreover, they had already found differences in the transcriptional mechanisms undergone by *soxT1A* and *soxT1B*. In this report, they extended their studies to the effects of SoxT1A and SoxT1B transporters on sulfur metabolism and regulatory systems.

This paper clearly shows the impact of sulfur transport on sulfur metabolism and its regulation, as well as the phylogenetic position of the targeted transporters. It is quite interesting that two similar transporters are independently involved in sulfur metabolism and signal regulation, respectively. This study is important not only for the regulatory mechanisms of sulfur metabolism, but also for examining the function of transporters in other metabolites. Therefore, I recommend that this paper deserves to be published in Communications Biology after minor revision.

We thank the reviewer for these positive comments.

1. I think that the *tsdA* mutant cannot metabolize thiosulfate to tetrathionate; however, I cannot completely understand why you used this mutant strain as a reference. Explaining the use of this mutant strain in the study support the reader's understanding.

In the section "Regulation of yeeE-like genes in Hyphomicrobium denitrificans" of the "Results" parts, we have introduced several sentences describing our reasons for using H. denitrificans Δ tsdA as a reference strain: "H. denitrificans has the genetic potential for two distinct pathways of thiosulfate oxidation, both of which occur or begin in the periplasm. At 5 mM, all thiosulfate is converted to tetrathionate. This reaction is catalysed by thiosulfate dehydrogenase (TsdA), a periplasmic diheme cytochrome c. At 2.5 mM or less, most thiosulfate is oxidized to sulfite and eventually to sulfate. However, the formation of tetrathionate is not completely suppressed in wildtype H. denitrificans even at low substrate concentrations. This makes phenotypic characterization of mutant strains difficult. Strains lacking thiosulfate dehydrogenase (Δ tsdA) are unable to form tetrathionate and are ideal for studying the processes involved in the complete oxidation of the substrate to sulfate. Therefore, all experiments reported in this and also our previous studies on the transcriptional regulation of sulfur oxidation were performed with a H. denitrificans Δ tsdA reference strain." Citations are omitted here but are provided in the manuscript.

2. The explanations of Fig. 4A and B in the text are reversed; therefore, either the explanations or the A, B in the figure need to be swapped.

The explanation in the text was rearranged.

3. The authors need to describe the phenotype of the *soxT1A* mutant near line 322, citing Figure S6.

This was done

Reviewer #2:

In this study, the authors identified two YeeE/YedE-family proteins occurred in the neighborhood of sulfur-metabolizing cluster in *Hyphomicrobium denitrificans*. Phylogenetic analysis assigned these two transmembrane proteins into SoxT1 group, term SoxT1A and SoxT1B. By measuring the transcript levels of marker genes and TS oxidation in the absence or presence of TS in Δ soxT1A or Δ soxT1B in combination with Δ soxR or Δ shdrR, the authors claimed that SoxT1A function as sulfur importer and SoxT1B as signal transducer. This is the central statement of this paper, however, there are many issues to be addressed.

Major comments:

1. Why use the Δ tsdA as reference strain? The background knowledge of Δ tsdA is needed.

*All three reviewers asked for background information on the use of *H. denitrificans* Δ tsdA. As mentioned above, we have included the missing explanation in the “Results” section.*

2. ‘The transcript abundance for *soxT1A* increased more than tenfold upon addition of thiosulfate in the *H. denitrificans* Δ tsdA reference strain, while *soxT1B* expression remained essentially unaffected’. There is no figure or table for this conclusion.

The data was published in a paper (Li et al 2023, Antioxidants). For the sake of clarity, we are quoting this publication one sentence earlier in the revised manuscript.

3. ‘A similar role may be played by YeiH in *H. denitrificans*’. The protein sequence similarity of YeiH should be provided to support this conclusion.

*An alignment of YeiH from *H. denitrificans* and *Paracoccus pantotrophus* is provided as Supplementary Figure 3 in the revised manuscript. We also included an overlay of the two protein structures predicted by Alphafold.*

4. The fold change expression of mRNA-Seq should be consistent, either log2 or not, for all described genes.

This was done.

5. ‘...the transcription of only three genes in the genomic sulfur oxidation region shown in Fig. 3a proved unaffected in the mRNA-Seq experiment...’. If so, the column of *shrR*, *soxT1B* and *soxR* in the presence of thiosulfate should be comparable to that in the absence of thiosulfate. In this figure, the fold change of these three genes in the presence of thiosulfate is near zero, meaning that they are far downregulated, rather than unaffected by thiosulfate. Please explain it.

*We thank the reviewer for noting our error. We inadvertently did not show a column for the three genes mentioned in the presence of thiosulfate. The *shdrR*, *soxT1B* and *soxR* genes were also transcribed under these conditions, but not significantly differently than in the absence of the sulfur compound. The figure was corrected accordingly.*

6. Fig. 3b, the transcript levels of *soxR* and *shrR* in the presence of thiosulfate are needed to support the conclusion that ‘...*soxT1B* expression, along with that of *soxR* and also *shdrR*, does not change significantly...’.

*We rephrased this part of the manuscript such that it becomes clear that RT-qPCR analyses for *soxR* and *shrR* in the presence of thiosulfate have already been published previously (Li et al 2023, Antioxidants). This work has been done for the Δ tsdA reference strain which is now explicitly stated in the sentence mentioned by the reviewer. RT-qPCR analyses with the*

regulator negative strains were performed with cells grown in the absence of thiosulfate as stated in the legend of Fig. 3b. In principle it would be desirable to quantify the transcript levels of the regulator negative strains not only in the absence but also in the presence of thiosulfate. However, both strains show very little growth during thiosulfate oxidation due to constitutive expression of all sox and shdr genes, resulting in massive formation of toxic sulfite (Li et al 2023, Antioxidants and BBA - Bioenergetics). It is virtually impossible to collect enough cell material for mRNA extraction during this growth phase.

7. Fig. 3b, the transcript levels of tested genes in the absence of thiosulfate in Δ tsdA Δ soxR and Δ tsdA Δ shdrR should be provided to draw this conclusion that ‘In the repressor-negative strains, soxT1A expression is high even in the absence of thiosulfate, while soxT1B transcript abundance is hardly affected’.

Figure 3b shows exactly what the reviewer asks for: both regulator-negative strains were grown in the absence of thiosulfate as indicated in the figure legend. As mentioned above, both strains constitutively oxidize thiosulfate but show very little growth while oxidizing the sulfur substrate. Substantial growth starts only after consumption of thiosulfate. This has been reported in Li et al 2023, Antioxidants and BBA - Bioenergetics.

8. Fig. 3b, why are the transcript levels of all tested genes in Δ tsdA Δ shdrR decreased compared to the reference strain Δ tsdA?

*We thank the reviewer for this remark. We added several sentences to the manuscript describing this observation and providing an explanation: “Note that the transcript levels for the tested genes are lower in the Δ tsdA Δ shdrR strain in the absence of thiosulfate than in the *H. denitrificans* Δ tsdA reference strain in the presence of the sulfur substrate. This observation cannot be fully explained yet but is certainly related to the second regulator SoxR. This protein is still present in the Δ tsdA Δ shdrR strain, and transcriptional repression is apparently not fully relieved by the absence of sHdrR alone. Experiments described further below confirm SoxR as the major regulator of sulfur oxidation in *H. denitrificans*. Clearly, the interplay of the two regulators needs to be investigated in more detail in the future.”*

9. Fig. 4a and Fig. S5b, assuming that SoxT1B function as signal transducer, the TS concentration should be decreased since the reaction of TS in the periplasm and the import system are intact. Please explain it.

In the strain lacking soxT1B, the genes for the Sox proteins including SoxT1A are not transcribed in the presence of thiosulfate, because the repressors are not released from the DNA. As a consequence, thiosulfate cannot be oxidized (see Fig. 5b).

10. How to exclude the possibility that the membrane protein serves as toxic product exporter? Please explain it.

If SoxT1A and/or SoxT1B served in export of sulfite, this toxic compound would accumulate in the cytoplasm in strains lacking the transporters and thus likely prevent growth completely. Just the opposite is observed, cells lacking either one of the transporters grow very well in the presence of thiosulfate and this is because it is not oxidized at all and the oxidation product sulfite never appears (Supplementary Figures 5 and 6).

11. Fig. 5 and Fig. 6, the fold changes of tested genes in the reference strain should be present for comparison.

We feel that this would be a redundant presentation of datasets. The full set of data for the reference strain is already provided in Fig 3b.

12. Fig. 6. assuming SoxT1A import certain sulfur compound derived from periplasmic TS oxidation, why is the transcript levels of *shdrA* and *soxXA* in $\Delta soxT1A$ far lower than that in $\Delta soxT1A \Delta soxR$ by TS addition, since the signal transducing pathway is intact?

It is true that the signal transduction cascade is still active in the mutant lacking SoxT1A. However, a complete and functional signal transduction cascade does apparently not result in extremely high transcription rates. Rather, transcription is tuned in such a way that it does not rise above levels that are sufficient to meet the metabolic needs. We added these considerations to the section describing the results obtained with $\Delta soxT1A$ Hyphomicrobium strains:

*“Remarkably, transcript level increases for *shdrA* and *soxXA* in *H. denitrificans* $\Delta tdsA \Delta soxT1A$ caused by the presence of thiosulfate appear low, 6.62 ± 0.13 and 2.92 ± 0.36 fold, respectively, when compared to the *soxT1A*-deficient strains that additionally lack either one of the studied regulator genes (Fig. 6). It is important to note that the increases determined for the $\Delta tdsA \Delta soxT1A$ strain are in the same range as observed for the $\Delta tdsA$ reference strain by RT-qPCR (20.11 ± 2.39 fold and 5.68 ± 1.18 fold for *shdrA* and *soxXA*, respectively, Fig. 3b) and in our mRNA-Seq experiments (19 and 6.7-8.9 fold, Supplementary Table 3). Increasing transcript abundance in this moderate range allows thiosulfate oxidation. However, if one of the repressors is missing, the whole system apparently loses its balance resulting in extremely high transcription levels (Fig. 6.) and thus energy-intensive biosynthesis of the sulfur-oxidizing system including a complete assembly machinery for its specific lipoate-binding proteins. Further studies are needed to elucidate how exactly the fine-tuning of transcription rates to metabolic needs is achieved in the reference strain, and whether transcriptional regulators other than SoxR and sHdrR are involved in this process. The latter seems likely, since the transcription of genes for LuxR, LysR, and Fur-type regulators is significantly affected by the presence of thiosulfate (Supplementary Table 3 and Supplementary Figure 3).”*

Why is the derepress effect of the knockout of *shdrR* on *shdrA* and *soxXA* much more significant in $\Delta soxT1A \Delta shdrR$ compared to that $\Delta soxT1B \Delta shdrR$ (Fig. 5)?

In $\Delta soxT1B \Delta shdrR$ and in $\Delta soxT1A \Delta shdrR$ the repressor SoxR is still present. This repressor apparently is the major regulator as stated in the manuscript. In the $\Delta soxT1A$ background, signal transduction to SoxR is still possible while this is not the case in the $\Delta soxT1B$ strain.

Why is the fold change value of *shdrA* so dramatically high in $\Delta soxT1B \Delta soxR$ in the presence of TS, unlike that in $\Delta soxT1A \Delta soxR$ in the same condition?

*In fact, the fold change for *shdrA* transcription is much higher in the $\Delta soxT1A \Delta soxR$ strain than in the $\Delta soxT1B \Delta soxR$ strain. At present, we cannot fully explain this finding but we assume that it is caused by a loss of fine tuning in the absence of either one of the repressors, so the text given above also applies to this comment.*

13. Fig. S6b, why did the TS concentration not decrease in $\Delta tdsA \Delta soxT1A$ if the substrate transported by SoxT1A is certainly not TS since effective oxidation of TS occurred in periplasm (in discussion part)?

Thiosulfate oxidation can only start in the periplasm when Sox proteins are made and active. As shown in Fig. 7 and also mentioned in the text, in Hyphomicrobium the Sox proteins release one thiosulfate sulfur atom as sulfate, the other is hooked up to the sulfur-binding protein SoxYZ. From there the sulfur has to be transported to the cytoplasm for further oxidation (Fig. 7). If sulfur transport into the cytoplasm is not possible due to the lack of a suitable import protein, i.e. SoxT1A, SoxYZ will be overloaded with sulfur after few reaction cycles and thiosulfate cannot be oxidized in substantial, measurable amounts.

Minor comments:

1. ‘...Serratia sp. ATCC39006...’ in the Introduction part, ‘sp.’ should be regular

Corrected

2. ‘Members of the YeeE/YeeD family...’ should be ‘Members of the YeeE/YedE family’

Corrected

3. ‘mRNA transcripts of 47 genes showed lower mRNA abundance than in its presence’ should be ‘mRNA transcripts of 47 genes showed lower mRNA abundance than in its presence’

Corrected

4. ‘A similar role may be played by YeiH ...’ should be ‘A similar role may be played by YeiH...’

Corrected

5. The positions of Fig. 4a and 4b should be switched.

The text was adapted (see above).

6. The y-axis of Fig. 6 should be changed into two-segments to show the value of shdrA.

We prefer to keep the figure as is.

Reviewer #3 (Remarks to the Author):

Li et al. have investigated the physical functions of SoxT1A and SoxT1B in *Hyphomicrobium denitrificans*. They observed some features that potentially explain how sulfur import into the cytoplasm and how regulators sense external sulfur, but the conclusion remains somewhat speculative. The work was not enough to prove the conclusion that SoxT1A delivers sulfur to the cytoplasm for its further oxidation and SoxT1B serves as a signal transduction unit for the transcriptional repressor SoxR. Below are concerns that should be addressed.

1. *H. denitrificans* contains many thiosulfate oxidation pathways. The authors mentioned that further thiosulfate oxidation steps take place in the cytoplasm, which requires the import of sulfane sulfur substrate and export of sulfite. The product of further oxidation in the cytoplasm is sulfite, which is also toxic to cell.

The following is now stated in the Results section:

“H. denitrificans has the genetic potential for two distinct pathways of thiosulfate oxidation, both of which occur or begin in the periplasm. At 5 mM, all thiosulfate is converted to tetrathionate. This reaction is catalysed by thiosulfate dehydrogenase (TsdA), a periplasmic diheme cytochrome c. At 2.5 mM or less, most thiosulfate is oxidized to sulfite and eventually to sulfate. However, the formation of tetrathionate is not completely suppressed even at low substrate concentrations. This makes phenotypic characterization of mutant strains difficult. Strains lacking thiosulfate dehydrogenase (Δ tsdA) are unable to form tetrathionate and are ideal for studying the processes involved in the complete oxidation of the substrate to sulfate.”

Why *H. denitrificans* chooses this risky pathway to oxidize thiosulfate rather than directly oxidizing thiosulfate in the periplasm by TsdA or Sox system.

As humans, we cannot fully understand evolution and why a bacterial genome collects the genetic capacity for several different, maybe even redundant pathways for the metabolism of a certain substrate. We have provided detailed considerations on this topic in a previous publication (Li et al 2023 BBA – Bioenergetics). One important aspect is that we should not only look at pure batch cultures of bacteria. Something like the formation of a toxic end product/intermediate might be detrimental in pure batch culture but advantageous in nature where the toxic intermediate might nurture other members of the microbial community.

I would recommend the authors clarify the using orders of these thiosulfate oxidation pathways in *H. denitrificans*. Whether the thiosulfate oxidation pathway in the cytoplasm is useless when thiosulfate could be oxidized in the periplasm by TsdA.

*We now provide information of the “using orders of thiosulfate oxidation pathways” (see above). We would also like to mention that oxidizing thiosulfate completely to sulfate instead of forming tetrathionate is by no means useless because by far more electrons can be fed into the respiratory chain when the substrate is fully oxidized. When a complete periplasmic Sox system is used, cytochrome *c* is the electron acceptor, while the cytoplasmic sHdr pathway enables direct reduction of NAD⁺, again an enormous advantage when we consider at which stage reducing equivalents can be fed into respiratory energy conservation.*

2. In Fig. 4, why the authors use Δ tsdA as the control rather than using WT strain? The growth curves and thiosulfate consumption rate of wild-type strain should be shown. Little was known about the differences of thiosulfate oxidation ability between WT and Δ tsdA strain.

We have provided all the necessary information in the paragraph given above and in previous publications. These are all cited in the current manuscript.

3. The author considered that Δ soxT1B may act as a signal transduction protein. However, the evidences for supporting this point are insufficient. The signal molecular transformed by membrane protein SoxT1B was failed to be identified.

It is true that we are not yet able to name the signal molecule transported by SoxT1B. Still, we consider our evidence sufficient to claim that SoxT1B is involved in signal transduction. This statement does not require the identification of the transported compound.

Why the sulfur oxidation substrates imported into the cytoplasm fail to serve as a sensing signal molecular for SoxR?

Recently, we published a detailed study on the properties of the SoxR repressor (Li et al 2023 Antioxidants). Electrophoretic mobility assays clearly showed that polysulfide serves as a signal molecule for SoxR. When incubated with polysulfide in a molar ratio of 1:1 an internal sulfur bridge is formed in SoxR, which causes a conformational change that leads to release of the repressor from its DNA binding sites. Thiosulfate also reacts with SoxR, but only at 50-fold excess over the protein. As stated above, the initial steps of thiosulfate degradation occur in the periplasm, and it is therefore unlikely that thiosulfate would ever reach concentrations in the cytoplasm that would be required to elicit a response from SoxR.

Why a specific transmembrane protein was applied only for signal transducing but not for sulfur oxidation substrate transforming?

We do not have a conclusive answer to this question. We assume that it is much easier for the cells to run two different modules, each dedicated to its specific purpose. Only a few protein molecules of SoxT1B have to be present to serve in a signaling cascade. On the other hand, cells must be rich in protein molecules that provide mass transport of sulfur as part of energy conservation. Separating the two functions has the advantage that large quantities of SoxT1A are only synthesized when they are really needed.

To confirm the function of SoxT1B, it is suggested that figure out which kind of sulfur compound serves as the signal of SoxR.

As pointed out above, polysulfide has already been identified as the signal for SoxR.

Purify SoxT1B and its mutant proteins and detect the affinity changes between the signal molecular and the proteins.

*This is of course an important question for the future. The work described in the current paper certainly does not solve all questions. We would like to emphasize that we are not working with *E. coli*. The construction of *Hyphomicrobium* mutants is very difficult and time-consuming. All methods for its genetic manipulation have been developed by our group. Of course, *in vitro* measurements with the pure SoxT1 proteins are highly desirable. On the other hand, it is not guaranteed that heterologous production of these membrane proteins in *E. coli* is straightforward. *E. coli* is a Gammaproteobacterium, *Hyphomicrobium* is an Alphaproteobacterium with different lipid composition of the cytoplasmic membrane. We will certainly try to overcome these problems but cannot provide this kind of data within the next year.*

Conduct molecular docking to reveal the mechanism how SoxT1B transfers this signal molecular and then figure out the signal binding pocket in SoxT1B.

*We performed a large number of docking runs with Autodock Vina. However, as had to be expected due to the small size of the probable tested substrates for the SoxT1 proteins (e.g. thiosulfate, disulfide, trisulfide, tetrasulfide, pentasulfide), it turned out to be impossible to draw any valid conclusions on specificity. Essentially all compounds are modelled to the same binding site, i.e. the same site, where thiosulfate is found in the *Spirochaeta thermophila* transporter.*

4. SoxT1A mutants are sulfur oxidation-negative despite high transcription levels of sulfur oxidation genes. The authors mentioned that SoxT1A delivers sulfur to the cytoplasm for its further oxidation. The only data to support this point comes from RT-qPCR analysis.

*We would like to point out that RT-qPCR analysis is not the only method that we applied. Our conclusions are also firmly based on the phenotypic analysis of the many *H. denitrificans* mutant strains constructed in the course of this work.*

To further confirm this conclusion, protein expression level of SoxXA and shdrA sulfur oxidation enzymes should be detected to exclude the post-transcriptional regulation in sulfur oxidation.

*In a previous work (Li et al 2023 BBA Bioenergetics), we have shown via Western blot that the sHdrA protein is much more abundant in the presence of thiosulfate than in its absence. Specific antibodies against any of the other sHdr or Sox proteins are not available. Still, we can exclude major post-transcriptional regulation. In 2018, we published a proteomic study comparing the *Hyphomicrobium* proteome on dimethyl sulfide (DMS) versus dimethyl amine (DMA) (Koch et al 2018 ISME Journal). Thiosulfate is an intermediate of DMS degradation. Accordingly, all proteins encoded in the sulfur oxidation region were much more abundant in DMS than in DMA grown cells, matching the transcriptome data presented in the current study.*

Detect whether the phenotype of Δ soxT1A mutant could be complemented by the genes responsible for thiosulfate import in other Alphaproteobacteria.

*We would like to point out that our work is novel. Thiosulfate import has yet not been elucidated in any other Alphaproteobacterium. The SoxT transporter of *Paracoccus pantotrophus* is likely to transport thiosulfate, but direct evidence for this proposal has not*

been provided yet. Therefore, suitable genes encoding proteins with exactly defined function for complementation of our Δ soxT1 mutant strains are not available at present.

Detect the affinity between SoxT1A and its transport substrate by isothermal titration calorimetry.

As stated above, we are currently not in the position to perform this type of experiment. Of course such studies are highly desirable in the future.

Construct conserved residue mutants of SoxT1A and detect their ability of thiosulfate oxidation.

*As stated above, methods for genetic manipulation of *H. denitrificans* are still rudimentary and time-consuming. We proved the essentiality of the same cysteine residues as in YeeE for SoxT1B and believe that this finding can be extrapolated with confidence to SoxT1A. We agree that it would be good to have the actual experimental data in the future.*

5. Page 9 line 250: as shown in Fig. 3b, --soxXA was selected as one of the four indicative genes-- According to the locus diagram, soxX and soxA are two genes and their expression levels were not the same as shown in Fig. 3a. soxXA labeled in the column diagram could be corrected as soxX or soxA.

SoxX and SoxA are two subunits of the heterodimeric protein SoxXA. Many papers have been published about the properties of this c-type heme binding protein. It has been isolated from a whole range of different sulfur oxidizers including Alphaproteobacteria. SoxA is the catalytic subunit, SoxX conducts away the released electrons. It is correct that the fold increases for the two genes are not exactly the same in our RNA-Seq analysis but they are certainly very similar and within the error range of the method. We therefore consider it appropriate to analyze their transcription in combination.

6. Page 9 line 253-254: In Fig. 3b legend ---the absence of thiosulfate are also shown. All changes are compared to *H. denitrificans* Δ tsdA in the absence of thiosulfate----, however, the corresponding results were missing in Fig. 3b.

The soxR and shdrR mutant strains were tested only in the absence of thiosulfate. Growth of these strains in the presence of thiosulfate is very slow and results cannot be obtained, see above.

7. In Fig.5b, Δ soxT1B, Δ soxT1BsoxR and Δ soxT1BshdrR should be corrected as Δ tsdA Δ soxT1B, Δ tsdA Δ soxT1BsoxR and Δ tsdA Δ soxT1BshdrR according to the description in Page10 line281.

Done

8. In Fig.6b, Δ soxT1A, Δ soxT1AsoxR and Δ soxT1AshdrR should be corrected as Δ tsdA Δ soxT1A, Δ tsdA Δ soxT1AsoxR and Δ tsdA Δ soxT1AshdrR according to the description in Page12 line320.

Done

9. In supply information Fig. 4b, Δ tsdA Δ soxT1B Δ shdrR should be corrected as Δ tsdA Δ soxT1A Δ shdrR.

Done

Manuscript ID.: **COMMSBIO-24-2898A**

Title: "Yee-like bacterial SoxT proteins mediate sulfur import for oxidation and signal transduction"

Corresponding Author: Christiane Dahl

All Authors: Jingjing Li, Fabienne Göbel, Hsun Yun Hsu, Julian Nikolaus Koch, Natalie Hager, Wanda Antonia Flegler, Tomohisa Sebastian Tanabe, Christiane Dahl

We thank the reviewers for the helpful comments. Please find our point by point answers below. For better readability our answers are printed in italics.

Reviewer #2 (Remarks to the Author):

This manuscript has been improved compared to the previous version. However, some issues are still not be addressed. Here are the specific comments.

1. Fig. 3b, Indications should be added in the RT-qPCR figure showing that the column of *ΔtsdA* reference strain is in the presence of thiosulfate and the columns of repressor absence strains are in the absence of thiosulfate.

The figure was modified as suggested.

2. Is *soxYZ* also regulated by SoxR?

Yes, soxYZ is also regulated by SoxR. This is evident from the mRNA-Seq data presented in Fig. 3a and has already been shown by RT-qPCR as reported in Li et al 2023, Antioxidants. In addition, we demonstrated that SoxR binds to the intergenic region between soxAX and soxYZ in vitro (Li et al 2023, Antioxidants).

3. Fig. 5 and Fig. 6. RT-qPCR of the reference strain is needed in each set of experiment since the absolute abundances of tested genes vary in different experiments. All the changes in Fig. 5 and Fig. 6 have to be calculated based on reference strain of the same set. Even though the response of the reference strain is similar to Fig. 3, the present of reference strain in each set has to be present to show the role of knocked out genes.

*RT-qPCR results for the *ΔtsdA* reference strain have now been added to the experiments depicted in Figs. 5 and 6.*

4. Fig. 6 The fold change of *shdrA* (~6-fold) is much lower in the presence of thiosulfate in *ΔtsdA ΔsoxT1A* than that in *ΔtsdA* (Fig. 3b, ~20 fold). They are definitely not in the same range. It did not make sense if the SoxT1A functions only in (transporting a) certain intermediate into cytoplasm. Also, since SoxR is the major regulator directly control the transcription of *sox* genes, the absence of SoxR should derepress the *shdrA* to the similar levels in both *ΔtsdA ΔsoxT1A ΔsoxR* and *ΔtsdA ΔsoxT1B ΔsoxR*. It is odd that the fold change of *shdrA* in *ΔtsdAΔsoxT1AΔsoxR* in the presence of thiosulfate is so high (>200 fold). Since the author could not provide more solid evidence to prove the transporter function of SoxT1A, it is not suitable to assert that SoxT1A as sulfur importer especially in the title.

We would like to point out that compared to the unphysiological fully de-repressed situation in strains deficient in one of the regulators in the presence of thiosulfate, where changes in

transcript abundance for *shdrA* range over 100-fold, a difference between ~6-fold and ~20-fold still seems to be in a similar range. It is not possible to derive function solely from changes in transcript abundances but the physiology of mutant strains as well as biochemical experiments have to be taken into account. All of the evidence we collected in the course of this study goes along with a function of SoxT1A in sulfur import. There are no results that contradict this conclusion. On the other hand, we acknowledge that we cannot completely rule out an involvement in the signaling process for this protein. This is now explicitly stated in the text at the end of the section "Role of SoxT1A in *H. denitrificans*".

We would furthermore like to point out to the reviewer, that our study focusses on the function of the membrane proteins SoxT1A and SoxT1B. In the course of phenotypic characterization of mutant strains, we unraveled the involvement of two different repressors in the overall process. We have clarified that there is an interaction between the repressors, but to elucidate all the details of this interplay is beyond the scope of our current study. We have stated that we collected evidence for SoxR being the major regulator, but we have not stated that this is an irrefutable fact. More work is clearly necessary to elucidate the details of SoxR-sHdrR interaction and the identification of possibly overlapping binding sites.

Reviewer #3 (Remarks to the Author):

This study focuses on investigating the function of SoxT1A and SoxT1B involved in sulfur metabolism and signal regulation in *H. denitrificans*. Although the authors emphasized that their study is novel, the methods for genetic manipulation of *H. denitrificans* are time-consuming and they are currently not in the position to perform some type of experiments, I still recommend that the experimental data as suggested in the review comments should be obtained before publishing this paper in Communications Biology.

1. The author mentioned that in the presence of thiosulfate, regulator negative strains including $\Delta tsdA \Delta soxR$ and $\Delta tsdA \Delta shdrR$ show very little growth during thiosulfate oxidation due to constitutive expression of all *sox* and *shdr* genes, and it is virtually impossible to collect enough cell for mRNA extraction during this growth phase. However, in $\Delta tsdA \Delta soxT1B \Delta soxR$, *sox* and *shdr* genes are also constitutive expression, and this strain also has thiosulfate oxidation ability, why its RT-qPCR analyses could be performed. I think these results are confused.

As requested by the reviewer, we now provide RT-qPCR results for the regulator negative strains $\Delta tsdA \Delta soxR$ and $\Delta tsdA \Delta shdrR$ grown in the presence of thiosulfate (Fig. 3c).

2. The authors considered that *H. denitrificans* SoxT1A is the only experimentally demonstrated sulfur importer in dissimilatory sulfur-oxidizing prokaryotes. However, which kinds of sulfur compound serving as the transport substrate transformed by SoxT1A failed to be identified in the manuscript. Whether these sulfur compounds could upregulate the expression of SoxT1A should be addressed. The proteins belonging to YeeE/YedE family are responsible for thiosulfate import was confirmed in *E. coli* previously, the findings that this family proteins have the similar function in *H. denitrificans* are not surprise. Although this work is interesting, breadth of discovery by the current experimental data is unlikely to result in major advance. Some efforts on the

model of how sulfur transported through the transporters should be added in this manuscript. Conducting some biochemical experiments about the interaction between the sulfur molecular and the transporters contribute to the understanding of functions of SoxT1A and SoxT1B in *H. denitrificans*.

We certainly agree with the reviewer that more experimental evidence would be desirable. The identification of the sulfur species transported through SoxT1A and SoxT1B is a major goal of future research. The reviewer's idea to test different possible transported sulfur species for their ability to induce transcription of the relevant genes is interesting, but would require a study of the toxic effects of hydrogen sulfide and polysulfide on the organism.